



# Performance of MAR (v3.11) in simulating the drifting-snow climate and surface mass balance of Adelie Land, East Antarctica

Charles Amory [1,2], Christoph Kittel [1], Louis Le Toumelin [2,3], Cécile Agosta [4], Alison Delhasse [1], Vincent Favier [2], and Xavier Fettweis [1]

[1]Department of Geography, UR SPHERES, University of Liège, Liège, Belgium
[2]Univ. Grenoble Alpes, CNRS, Institut des Géosciences de l'Environnement, Grenoble, France
[3]Univ. Grenoble Alpes, Université de Toulouse, Météo-France, CNRS, CNRM, Centre d'Études de la Neige, Grenoble, France
[4]Laboratoire des Sciences du Climat et de l'Environnement, LSCE-IPSL, CEA-CNRS-UVSQ, Université Paris-Saclay, Gif-sur-Yvette, France

**Correspondence:** C. Amory (charles.amory@univ-grenoble-alpes.fr)

**Abstract.** Drifting snow, or the wind-driven transport of snow particles and their concurrent sublimation, is a poorly documented process on the Antarctic ice sheet, inherently lacking in most climate models. Since drifting snow mostly results from erosion of surface particles, a comprehensive evaluation of this process in climate models requires a concurrent assessment of simulated drifting-snow transport and the surface mass balance (SMB). In this paper a new version of the drifting-snow scheme currently embedded in the regional climate model MAR (v3.11) is extensively described. Several important modifications relative to previous version have been implemented and include notably a parameterisation for drifting-snow compaction, differentiated snow density at deposition between precipitation and drifting snow, and a rewriting of the threshold friction velocity for snow erosion. Model results at high resolution (10 km) over Adelie Land, East Antarctica, for the period 2004-2018 are presented and evaluated against available near-surface meteorological observations at half-hourly resolution and annual SMB estimates. MAR resolves the local drifting-snow frequency and transport up the scale of the drifting-snow event and captures the resulting observed climate and SMB variability. This suggests that this model version can be used for continent-wide applications, and that the approach of drifting-snow physics as proposed in MAR can serve as a basis for implementation in earth system models.

## 1 Introduction

A significant portion of the surface area of Antarctica is affected by wind-driven ablation. The net snow accumulation at the ice-sheet surface, i.e. the surface mass balance (SMB) is the resultant of mass gains (precipitation and riming), mass losses (water runoff and surface sublimation) and wind-driven snow transport (or the horizontal advection of precipitating and eroded snow particles by wind) which can either result in mass gain (deposition) or loss (erosion). The snow mass sublimated during transport is lost by the ice-sheet surface to the atmosphere when particles originate from the surface. Although wind-driven snow sublimation has most often been described as an independent term in the SMB equation (e.g., van de Berg et al., 2006; Lenaerts et al., 2019), its contribution to surface mass loss is inherently included in the spatially integrated erosion/deposition





balance. Drifting and blowing snow are usually conventionally distinguished as the wind-driven transport of snow particles respectively below and above a height of 2 m above ground. In this study both processes are combined into the single denomination of drifting snow for convenience, and erosion, deposition, transport of wind-driven snow particles and their concurrent

sublimation are all referred to as drifting-snow processes.

The net erosion/deposition balance in areas subject to drifting snow is mainly governed by the interactions between the complex ice-surface topography, near-surface flow and surface snowpack state. Katabatic winds flowing over the surface of the Antarctic ice sheet accelerate down steep surface slopes, causing erosion when the wind shear stress is high enough to dislodge particles from the surface. Wind redistribution of snow occurs more generally on every spatial scale at which topo-

graphic features generate acceleration or deceleration of the near-surface flow, enhancing or reducing local SMB gradients up to subkilometre scales (Agosta et al., 2012; Dattler et al., 2019; Kausch et al., 2020) and occasionally forming extensive areas of near-zero to negative SMB in windy and dry interior regions of the ice sheet (Bintanja, 1999; Scambos et al., 2012). From a modelling perspective, this means that drifting-snow processes vary as a function of the horizontal grid size (Lenaerts et al., 2012b). This is of particular importance for resolving the spatial variability in SMB at the ice-sheet margins where

drifting-snow processes can be major components of the local SMB (King et al., 2004; Gallée et al., 2005; Frezzotti et al., 2007; Lenaerts and van den Broeke, 2012), resulting in the transport and sublimation of large volumes of snow and/or their export off the continent boundaries (Scarchilli et al., 2010; Palm et al., 2017).

As drifting snow triggers interactions between the atmosphere and the ice-sheet surface, a comprehensive evaluation of drifting-snow processes using snow-transport models requires consistency between model results and observations for both

mass transport and net accumulation rates simultaneously. The much larger availability of SMB observations compared to drifting-snow measurements in Antarctica and the wider applications of modelled SMB products have led model development and evaluation exercises to focus primarily on the representation of the SMB, with a secondary or most often non-existent consideration for drifting-snow processes (Lenaerts et al., 2019; Mottram et al., 2020). SMB and drifting-snow transport, however, are not independent variables. Arbitrary adjustments of model parameters favouring one can be made at the expense

of the other (e.g., van Wessem et al., 2018). Similarly, the neglection or underestimation of drifting-snow processes induces a smoothing of the modelled SMB gradients across areas of complex topography (Agosta et al., 2019), and can ultimately lead to overestimation of the snow mass input in regionally integrated SMB calculations (Frezzotti et al., 2004; Das et al., 2013).

Measurements of drifting-snow mass fluxes are particularly interesting for evaluating snow-transport models since they constitute the integrated result of all the feedback and dynamical mechanisms (i.e., precipitation, local erosion, horizontal

advection from upwind areas, and sublimation) that contribute to the presence, amount, and residence time of snow particles in the air. The general scarcity in drifting-snow measurements in polar regions is however a constraint to the development of parameterisation schemes for large-scale applications and currently hinders quantitative evaluations of contrasting, continent-wide model estimates of drifting-snow mass and sublimation fluxes in Antarctica (see Lenaerts and van den Broeke, 2012; Palm et al., 2017; van Wessem et al., 2018; Agosta et al., 2019). Numerical challenges associated with modelling drifting snow

at the regional scale also arise from the non-linearity of drifting-snow processes and from their numerous interactions with the atmosphere and the snow surface organized in a complex system of positive and negative feedback mechanisms (Lenaerts and





van den Broeke, 2012; Gallée et al., 2013). This is mirrored through a high sensitivity of model results to parameter choices and significant discrepancies between simulated and observed snow mass fluxes (Lenaerts et al., 2014; Amory et al., 2015; van Wessem et al., 2018).

The polar-oriented regional climate model MAR includes a drifting-snow scheme initially developed to improve the representation of the Antarctic SMB (Gallée et al., 2001). However, the drifting-snow scheme has only been used so far to study separately wind-driven ablation (Gallée, 1998; Gallée et al., 2001, 2005) and individual drifting-snow events (Gallée et al., 2013; Amory et al., 2015) in coastal East Antarctica. Former physical parametrisations of drifting snow in MAR did not lead to a continent-wide agreement between model simulations and both drifting-snow and SMB observations. As a result, drifting

snow has been kept disabled in recent decades-long investigations of the SMB of the Greenland (e.g., Fettweis et al., 2017, 2020) and Antarctic (e.g., Kittel et al., 2018, 2020; Agosta et al., 2019) ice sheets with MAR, despite the potentially missing aspects related to the important feedback processes induced by drifting snow.

In this paper a modified version of the original drifting-snow scheme implemented in MAR is assessed through a concurrent evaluation of the drifting-snow climate and SMB reproduced by the model against a multi-year database of drifting-snow mass

fluxes and SMB estimates. The evaluation focuses on the marginal slopes of Adelie Land, a katabatic-wind region of East Antarctica which experiences drifting snow frequently (Amory, 2020a) and where the SMB exhibits a high spatial variability related to drifting-snow processes (Agosta et al., 2012). The coupled atmospheric and snowpack components of MAR are presented in Sect. 2 and the drifting-snow scheme is described in Sect. 3 together with the new developments and main changes relative to the original version. Section 4 provides information on the study area, the available data and the evaluation strategy.

The modelled near-surface climate, drifting-snow frequency, mass transport, and SMB are evaluated in Sect. 5. The sensitivity to the model version and input parameters of the drifting-snow scheme are discussed in Sect. 6 before concluding the paper.

## 2  Model descriptions

### 2.1  Atmospheric model

MAR is a hydrostatic atmospheric model originally developed to simulate the climate over high-latitude regions. The atmo-

spheric dynamics in MAR are described in Gallée and Schayes (1994). Cloud microphysical processes and resulting precipitation are simulated by solving conservation equations for specific humidity, cloud droplets and ice crystals, raindrops and snow particles (Gallée, 1995; Gallée et al., 2001). The radiative transfer through the atmosphere is adapted from Morcrette (2002), and cloud radiative properties are computed from the concentration of cloud droplets and cloud ice crystals. Turbulence is resolved in the surface layer following the Monin-Obukhov similarity theory, and above the surface layer using a local closure

scheme adapted to the stable boundary layer (Gallée et al., 2015).

In this study MAR version 3.11 in its Antarctic set-up (Agosta et al., 2019) is used with the updates described in (Kittel et al., 2020), simply referred to as MAR hereafter. The simulations are performed on a grid of $80 \times 80$ cells at 10 km horizontal resolution to reduce computational cost and facilitate development and sensitivity experiments. The time step is set to 60 s. The topography is obtained through aggregation of the 1 km Bedmap2 surface elevation dataset (Fretwell et al., 2013). The





model was run over the period 2004–2018 and forced at its lateral boundaries (pressure, wind speed, temperature, specific humidity), at the top of the troposphere (temperature, wind speed) and at the ocean surface (sea ice concentration, sea surface temperature) by 6-hourly ERA5 reanalysis fields (Hersbach et al., 2020). The atmosphere is described on a stretched grid with 24 vertical terrain-following levels, of which 8 and 5 are respectively located in the lowest 100 and 20 m of the atmosphere with a lowest level at 2 m. The model is relaxed towards the forcing solutions of wind speed and temperature from the top

of the troposphere (i.e., above 10 km) to the uppermost atmospheric level following (van de Berg and Medley, 2016). The snowpack was uniformly initialized with a density of 500 kg m$^{-3}$, and the model was first run from 1994 so that the snowpack had reached equilibrium with the climate preceding the period of interest.

## 2.2 Snowpack model

The atmospheric part of MAR is coupled with the one-dimensional multi-layer surface model SISVAT (Soil-Ice-Snow-Vegetation-

Atmosphere-Transfer) which handles energy and mass transfer between the surface and the atmospheric boundary layer (De Ridder and Gallée, 1998). SISVAT includes a representation of snow (Gallée and Duynkerke, 1997; Gallée et al., 2001) and ice (Lefebre et al., 2003) layers and comprises subroutines for snow metamorphism, surface albedo, meltwater percolation, retention and refreezing. In the present study, 30 snow/ice layers are used to describe the snowpack with a fixed total resolved snowpack of 20 m in thickness. An aggregation scheme automatically manages the stratification of the snowpack due

to precipitation, erosion/deposition of snow, mechanical compaction, thermal and melting/refreezing metamorphism, enabling a dynamical evolution of the physical characteristics (temperature, density, water content, grain shape and size) of the different layers over time. If precipitation or deposition occurs when the snowpack already comprises the maximum number of layers, the formation of a new layer at the surface is achieved through aggregation of internal sub-surface layers. More generally, aggregation of adjacent layers is permitted according to the similarity of their physical properties. Thick layers can also be split

to refine the discretisation of the snowpack when the number of layers is lower than 10. Maximum layer thicknesses of the 4 uppermost layers of the snowpack are also prescribed (0.02, 0.05, 0.1, and 0.3 m) to ensure a fine discretisation adapted to the description of sub-surface processes such as heat exchange with the surface and diffusion within the snowpack. Mass and heat are conserved along the snowpack stratification procedure.

## 3 Drifting-snow scheme

### 3.1 Initiation of drifting snow

Erosion of snow is usually considered to initiate when the shear stress exerted by the flow at the surface (determined by the friction velocity $u_*$ in m s$^{-1}$) exceeds the threshold value for aerodynamic entrainment, i.e., the threshold friction velocity $u_{*t}$ (in m s$^{-1}$) determined by the resistive gravitational and cohesive forces. Resistive forces depend on temperature (Schmidt, 1980) and metamorphism history (Gallée et al., 2001) of the snow surface and involve various snow particle characteristics

such as inter-particle cohesion, density, grain shape and size. It follows that an accurate prognostic of $u_{*t}$ requires a detailed



representation of these characteristics, which are particularly undocumented in Antarctica and for which measurements are generally very limited in the literature. As an alternative, density has often been proposed as a governing factor in parameterisations of $u_{*t}$ (e.g., Liston et al., 2007; Lenaerts et al., 2012a). The same approach is followed here; erosion in the model occurs when $u_* > u_{*t}$, where $u_{*t}$ is imposed by the uppermost snow layer density

$$u_{*t} = u_{*t_0} exp(\frac{\rho_i}{\rho_0} - \frac{\rho_i}{\rho_s}), \tag{1}$$

in which $\rho_s$ is the surface snow density (kg m$^{-3}$), $\rho_i$ is the density of ice (920 kg m$^{-3}$) and $\rho_0$ is the density of fresh snow (set to 300 kg m$^{-3}$). The expression for $u_{*t_0}$ is retained from (Gallée et al., 2001)

$$u_{*t_0} = \frac{log(2.868) - log(1 + i_{ER})}{0.085} C_D^{0.5}, \tag{2}$$

$$i_{ER} = 0.75d - 0.5s + 0.5 \tag{3}$$

$$C_D = \frac{u_*^2}{U^2} \tag{4}$$

where $i_{ER}$ is an erodibility index describing the potential for snow erosion, $d$ (dendricity) and $s$ (sphericity) are the snow grain shape parameters, $C_D$ the drag coefficient for momentum, $U$ (m s$^{-1}$) the wind speed in the lowest model and $u_*$ is classically obtained through integration of stability correction functions for momentum over the atmospheric boundary layer. Dendricity represents the remaining initial geometry of fresh snow particles, and varies from 0 to 1 with high values of $d$ describing fresh snow layers. Sphericity varies equally from 0 to 1 and defines the ratio of rounded to angular shapes in the snow layer. In previous studies involving drifting-snow applications with MAR, $i_{ER}$ was defined as a function of surface snow characteristics ($d$, $s$ and grain size). To reduce the number of sensitivity parameters, as in Lenaerts et al. (2012a) $u_{*t}$ is assumed independent on particle size and constant snow grain shape parameters are assigned ($d = s = 0.5$), implying an erodibility index of 0.625 and a minimum $u_{*t}$ value of 0.3 m s$^{-1}$ for $\rho_s$ of 300 kg m$^{-3}$. Additional criterion for snow erosion requires that $\rho_s$ does not exceed $\rho_{max} = 450$ kg m$^{-3}$.

### 3.2 Snow transport modes

Particle motions in drifting snow are generally described through the two main transport modes consisting in saltation and turbulent suspension. Once the resistive forces have been overcome by the atmospheric drag force, erosion initiates through the saltation process, in which particles become mobile and periodically bounce on the surface within heights of the order of 10 centimetres. Turbulent suspension of snow occurs when snow particles obtain sufficient upward momentum to be entrained in the atmosphere by turbulent eddies from the top of the saltation layer without contact with the surface.

The drifting-snow scheme of MAR uses a set of semi-empirical formulations to predict the contribution of steady-state saltation and turbulent suspension to the airborne snow mass. In the model the particle ratio in the saltation layer $q_{salt}$ (kg





kg$^{-1}$ ; mass of saltating snow particles per unit mass of atmosphere) is parameterised as a function of the excess of shear stress responsible for removal of snow particles from the surface following the expression of Bintanja (2000) derived from Pomeroy and Gray (1990)

$$q_{salt} = e_{salt}\frac{u_*^2 - u_{*t}^2}{gh_{salt}},$$

(5)

with $e_{salt}$ = 1/(3.25$u_*$) the saltation efficiency, $g$ the gravitational acceleration (m s$^{-2}$), and $h_{salt}$=0.08436$u_*^{1.27}$ the thickness of the saltation layer (m) as proposed by Pomeroy and Male (1992). The saltating particle ratio is considered as constant throughout the saltation layer and serves as a lower boundary condition for the suspension layer.

The formulation of $q_{salt}$ is given for stationary conditions. Although the non-linear relationship between the flow and snow mass flux can induce fluctuations in particle concentration in non-stationary-conditions (Aksamit and Pomeroy, 2018), numer-

ical simulations suggest that steady-state saltation is achieved within an interval of a few seconds (Nemoto and Nishimura, 2004; Huang et al., 2016), well below the model time step of 60 s. With wind speeds at the surface typically reaching 10 m s$^{-1}$ during drifting-snow occurrences (Fig. S3), this corresponds to characteristic lengths of a few tens of metres, i.e. 3 orders of magnitude lower than the horizontal resolution of 10 km used in this study, indicating that a formulation of a stationary particle ratio remains appropriate in this context.

Snow transport in turbulent suspension is computed by the tri-dimensional turbulence and advection schemes of MAR, enabling a discretisation of snow transport profiles along the vertical grid of the model. The suspension layer receives contributions from snowfall and drifting-snow particles advected from upwind grid cells (top and lateral influx). Snow particles are suspended in the first model level through diffusion from the saltation layer (bottom influx). The surface (upward) turbulent flux of snow particles $u_*q_{s*}$ (m s$^{-1}$ kg kg$^{-1}$) is expressed assuming that it follows a bulk flux formulation

$$u_*q_{s*} = C_D U \zeta (q_s - q_{salt}),$$

(6)

where $q_{s*}$ (kg kg$^{-1}$) is the turbulent scale for snow particles, $\zeta$ is the ratio of eddy diffusivities for suspended particles and momentum and $q_s$ is the snow particle ratio (kg kg$^{-1}$) taken at the lowest model level. Because of fragmentation upon repeated collision between each other and with the snow surface, drifting-snow particles have smaller radius, and thus have smaller settling velocities than snow particles that have not yet experienced contact with the surface (Bintanja, 2000). As MAR cur-

rently does not distinguish snow particles originating from the surface from those directly formed by its cloud microphysics, the factor $\zeta$ was introduced in the original formulation of Gallée et al. (2001) to enhance the upward turbulent particle transport and compensate for a likely overestimation of the settling velocity of drifting-snow particles (Gallée et al., 2005). Following Bintanja (2000), $\zeta$ is set to 3.

### 3.3   Interactions with the atmosphere

Eroded snow is transmitted to the atmosphere by the surface scheme and added to the pre-existing airborne snow mass without distinction on the source of particles. Thermodynamic interactions of airborne snow particles with the atmosphere are handled by the cloud microphysical scheme. The increase in air density due to the presence of snow particles is taken into account by




modifying the formulation of virtual potential temperature (Gallée et al., 2001). Sublimation of snow particles occurs along their residence into the atmosphere and is parameterised as a function of undersaturation of air (Lin et al., 1983). The latent

heat consumption and humidity release caused by airborne-snow sublimation are directly accounted for in the energy and mass budget of each atmospheric layer in which sublimation occurs. This ensures that the model captures i) the negative feedback of sublimation through the increase in relative humidity of the air, ii) advective transport of humidity, and iii) the sublimation-induced cooling increasing air density and inhibiting upward turbulent motions (Bintanja, 2001). Weakening of drifting snow in response to decreasing $u_*$ as turbulence declines is reflected through the dependency of the surface snow turbulent flux $u_* q_{s*}$

on the difference $u_*^2 - u_{*t}^2$.

Airborne snow particles interact with the radiative transfer through the atmosphere and affect the surface energy budget by modulating downwelling irradiance, similarly to optically thin, low-level clouds (Yamanouchi and Kawaguchi, 1984; Mahesh et al., 2003; Le Toumelin. et al., 2020). Representing the radiative contribution of drifting-snow clouds is achieved in the model by including the snow particle ratio $q_s$ in the computation of cloud optical depth and emissivity (Gallée and Gorodetskaya,

195   2010).

### 3.4   Interactions with the surface

Alteration of surface characteristics through erosion/deposition of snow influences in turn the occurrence of drifting snow through various feedback mechanisms. Snow deposited at the surface during drifting snow is subject to the combined actions of wind and saltation which break original crystal shapes and favour the formation of smaller, rounded snow grains (Sato

et al., 2008), leading to enhanced sintering, more efficient mechanical packing and increased density (Vionnet et al., 2013; Sommer et al., 2018). Drifting-snow compaction, together with the exposure of denser snow or ice layers through erosion and/or sublimation, both naturally contribute to reduce the likelihood of additional drifting snow. The erosion/deposition process also influences the surface energy budget by modifying the surface albedo, which largely determines the energy available for melting. Surface melting reduces or even inhibits the potential for erosion in summer by increasing water content, density and

cohesion (Li and Pomeroy, 1997). Capturing these effects is thus of particular significance to account for temporal variations in drifting-snow frequency over peripheral regions of the Antarctic ice sheet (Lenaerts and van den Broeke, 2012). A different feature of the current drifting-snow scheme of MAR contrasting with earlier versions is that, instead of being distributed over several upper snow layers, the influence of erosion and deposition at each time step is restricted to the uppermost snow layer only. Snow layers with different characteristics may thus be deposited or exposed successively at the top of the snowpack

during a drifting-snow event, thus influencing the simulated surface albedo.

The current version uses fixed values for the characteristics of deposited snow but implicitly accounts for differences between snowfall and eroded particles. The characteristics of fresh snow ($\rho_0$, $d = 1$ and $s = 0$) differ from those of drifting-snow particles which are assumed to have completely lost their initial shape through collision and sublimation during transport and to be fully rounded ($d = 0$ and $s = 1$), although numerical simulations suggest the coexistence of various particle shapes during fully

developed drifting snow (Huang et al., 2011). Drifting snow is deposited with a density $\rho_{DR}$ assumed to be that of the current surface layer, with the restriction that $\rho_{DR}$ does not exceed $\rho_{max}$ to account for maximum surface snow density values observed



in Antarctica (Agosta et al., 2019) and allow for deposition of snow over more compacted snow and/or ice surfaces. The surface density $\rho_s$ is updated according to a relative contribution of both types of particles

$$\rho_s = \rho_0(1 - f_{DR}) + \rho_{DR}f_{DR}, \tag{7}$$


$$f_{DR} = 1 - \frac{q_{s,z_{lim}}}{q_s}, \tag{8}$$

where $f_{DR}$ is the drifting-snow fraction varying between 0 and 1 and $q_{s,z_{lim}}$ is the snow particle ratio at the atmospheric level closest to $z_{lim}$ (m) where the contribution of drifting-snow particles to the mass ratio is assumed to be negligible compared to the contribution of snowfall. A value of 100 m above surface has been adopted for $z_{lim}$ in accordance with the average depth

of drifting-snow layers over Antarctica as retrieved from remote sensing techniques (Mahesh et al., 2003; Gossart et al., 2017; Palm et al., 2018). Snowfall conditions imply $q_{s,z_{lim}} \sim q_s$ , low values of $f_{DR}$ and snow is deposited at the surface with a predominant contribution of $\rho_0$. Conversely, drift conditions imply $q_{s,z_{lim}} << q_s$ , high values of $f_{DR}$ and $\rho_s$ tends towards $\rho_{DR}$.

The influence of post-depositional processes is parametrised by increasing the density of the uppermost snowpack layer

in each grid cell in which erosion occurs. The temporal evolution of surface density along the range of values for which snow remains erodible is parameterised according to a linear densification rate from the fresh snow value $\rho_0$ (assumed to be representative of snow that have been barely altered by post-depositional processes) to the prohibitive density value for snow erosion $\rho_{max}$, i.e.

$$\frac{d\rho_s}{dt} = \frac{\rho_{max} - \rho_0}{\tau_{DR}}, \tag{9}$$

in which the characteristic time scale for drifting-snow compaction $\tau_{DR}$ is set to 24 h. This value corresponds to the average duration of drifting-snow events reported in (Amory, 2020a) and is used here as the typical duration for exhaustion of erodible snow to be reached. The linear behavior of the densification rate follows the linear increase in surface snow density retrieved from measurements performed during a drifting-snow event in Adelie Land (Fig. S2). Further details on this experiment are provided in supplementary materials (Sect. S1). Since $u_{*t}$ is parametrised as an increasing function of $\rho_s$, Eq.(9) prevents large

(positive) values of the difference $u_* - u_{*t}$ to endure through time and thus acts as a negative feedback for snow erosion.

### 3.5 Erosion

For each continental grid cell MAR calculates the actual snow mass eroded from the snowpack $ER$ (kg m$^{-2}$) during the current time step according to the following chain of events:

1. The snow particle ratio $q_s$ in the first model level, which includes the contributions of snowfall, airborne-snow subli-
mation and advection of snow as computed by the cloud microphysical and turbulence schemes, is transmitted to the surface scheme.



2. A potential maximum erosion $ER_{max}$ (kg m$^{-2}$) is estimated from the surface turbulent flux of snow particles $u_* q_{s*}$ computed from step 7 at the previous time step: $ER_{max} = \rho_a u_* q_{s*} dt$.

3. Actual erosion ER is calculated from removal of snow from the surface snowpack layer until ERmax is reached or the layer has been entirely eroded.

4. Snow at the surface densifies following Eq. (9).

5. The drift fraction is obtained from Eq. (8). The snowpack structure is adjusted to account for snow deposition at the surface according to Eq. (7).

6. The threshold friction velocity $u_{*t}$ and the saltation particle ratio $q_{salt}$ are deduced from Eqs. (1) and (5).

7. The surface turbulent flux of snow particles $u_* q_{s*}$ is computed from Eq. (6).

8. The contribution of erosion is added to $q_s$, which is then transmitted to the turbulence, cloud microphysical and radiative schemes to compute advection and interactions of airborne snow with the atmosphere.

### 3.6  Surface roughness

Drifting snow is responsible for the development of surface microrelief, whose spatial arrangement combined to the orientation of the wind determines the roughness length for momentum $z_0$. Because of wind-driven reshaping of the snow topography and the diversity in surface types, $z_0$ varies by several orders of magnitude with time and space across the Antarctic continent (Amory et al., 2017). The sensitivity of MAR to the parameterisation of surface roughness has been discussed in Amory et al. (2015) who demonstrated that a dynamic representation of $z_0$ is required to improve the modelling of wind speed and drifting-snow fluxes in Adelie Land. Due to inconsistencies between observed and modelled temporal variations in $z_0$ values, the former parameterisation of $z_0$ was changed and is now computed as a function of the air temperature only (Amory et al., 2017). This parameterisation was developed so that $z_0$ fits the observed seasonal variations between high ($> 10^{-3}$ m) summer and lower winter values in coastal Adelie Land, for air temperatures above -20 °C. For lower temperatures, a constant $z_0$ value of $2 \, 10^{-4}$ m is set in agreement with observations on the Antarctic plateau (Vignon et al., 2017).

Roughness features alter the spatial distribution of wind shear near the surface through pressure fluctuation gradients in their immediate vicinity (i.e., the form drag). This drag partitioning results in a loss of momentum from the near-surface flow by turbulent friction, which in turn reduces the energy budget available for erosion in the form of negative feedback. Previous versions of the drifting-snow scheme in MAR included a parameterisation of drag partitioning developed for non-erodible roughness elements encountered in desert-like environments (Marticorena and Bergametti, 1995). While snow roughness features have been shown to effectively exert significant form drag inhibiting snow erosion, this mainly occurs for near-surface flows and roughness features of crosswise orientations and can essentially vanish through a rapid streamlining process of the microrelief under erosive conditions (Andreas and Claffey, 1995; Amory et al., 2016). Sensitivity experiments revealed that the drag partition scheme was responsible for a strong inhibiting effect on snow erosion beyond the observed magnitude of





**Table 1.** Main characteristics of the two measurement locations used for the evaluation of MAR.

| Station | Location | Elevation (m) | Elevation bias (m) | Observation period |
|---|---|---|---|---|
| D47 | 67.4°S, 138.7° | 1560 | -8 | Jan. 2010 – Dec. 2012 |
| D17 | 66.7°S, 139.9° | 450 | -66 | Feb. 2010 – Dec. 2018* |

*Station still operative.

the negative feedback mechanism, possibly as it does not account for the dynamical and erodible nature of snow microrelief (Amory et al., 2015). Consequently, it has been disabled in the current model version.

## 4 Field area, observation data and evaluation methods

The near-surface climate in coastal Adelie Land is dominated by strong katabatic flows which drain cold air from the continental interior toward the steep coastal escarpment, enabling the regular incidence of well-developed drifting-snow events throughout the year (Amory, 2020a). High erosion rates and export of drifting snow combine to melt and sublimation to produce local net ablation at the surface and resulting persistent blue-ice areas near the coast on the steepest part of the ice margin (Genthon et al., 2007; Favier et al., 2011). In a fairly narrow transition, net accumulation is observed further inland despite significant drifting snow (Barral et al., 2014; Amory et al., 2017), with annual SMB values displaying a high, kilometre-scale variability as a result of wind redistribution (Agosta et al., 2012).

The performance of MAR in reproducing the drifting-snow climate of Adelie Land is evaluated against 2-m wind speed and direction, air temperature and air relative humidity observations collected at two locations 100 km apart, D47 and D17 (Fig. 1, Table 1). Data at half-hourly intervals are available from automatic weather stations installed at both sites and operated by the Institut des Géosciences de l'Environnement (IGE) over the periods 2010-2012 for D47 and 2010-2018 for D17 (Amory et al., 2020). All datasets are reported as quality-controlled. At each station relative humidity is originally given with respect to water, and has been converted to be expressed with respect to ice in subfreezing conditions following Goff and Gratch (1945) formulae and using the 2-m temperature record in the conversion. Climate variables extracted from the lowest model level (2 m) and the nearest grid cell to the observation location are used for comparison.

Half-hourly meteorological data also include drifting-snow mass fluxes measured almost continuously over their respective observation periods using acoustic second-generation FlowCapt™ sensors (hereafter referred to as 2G-FlowCapt™). The instrument consists of a 1 m long tube containing electroacoustic transducers that convert the acoustic vibration caused by the drifting-snow particles colliding with the tube into a snow mass flux integrated over the exposed length of the tube. At D47, two 2G-FlowCapt™ sensors were installed and superimposed vertically to sample the first two meters above ground and detect the onset of drifting snow. Site D17 was initially equipped with only one 2G-FlowCapt™ set up close to the surface, and completed with a second instrument in December 2012 to match the configuration of D47. As precipitating snow particles directly originating from clouds cannot be discriminated from saltating and/or suspended snow particles relocated from the





ground, measured snow mass fluxes account for all forms of wind-driven snow along the sampling height. The IGE database

then enables the evaluation of simulated drifting snow and the relative climate against time-averaged measurements of bulk flow and mass flux conditions, consistently with the steady-state drifting-snow physics implemented in MAR. An extensive description of the drifting-snow and meteorological equipment at D47 and D17 can be found in Amory (2020a).

Thorough evaluation of drifting snow requires consistency between observed and modelled drifting-snow mass fluxes. Near-surface snow mass fluxes simulated by MAR can be inferred from $q_s$ at the lowest vertical model level (2 m) which, similarly

to the 2G-FlowCapt™, does not distinguish on the origin of particles. By approximating the mean speed of suspended particles with the mean wind speed $U$ at the lowest model level, an average, horizontal (vertically integrated), near-surface drifting-snow mass flux $\mu_{MAR}$ (kg m$^{-2}$ s$^{-1}$) can be expressed as

$$\mu_{MAR} = U\rho_a q_s, \tag{10}$$

where $\rho_a$ is the air density (kg m$^{-3}$). With a lowest level at 2 m height, MAR does not capture the strong exponential decrease

in snow mass flux with height existing close to the surface (e.g., Mann et al., 2000; Nishimura and Nemoto, 2005). As the snow particle ratio $q_s$ has the same value throughout the model level, comparison between model and observations is performed by combining, when available, snow mass flux estimates at both measurement levels into an average, near-surface, drifting-snow mass flux $\mu_{OBS}$ (kg m$^{-2}$ s$^{-1}$) calculated through

$$\mu_{OBS} = \frac{\mu_1 h_1 + \mu_2 h_2}{h_1 + h_2} \tag{11}$$

where $\mu_i$ is the observed snow mass flux integrated over the exposed length $h_i$ of the corresponding 2G-FlowCapt™ sensor (Amory et al., 2015).

Modelled SMB is compared with observations obtained from annual measurements of 91 snow stakes distributed every ∼1.5 km along a 150 km long transect (see Agosta et al. (2012) for description) that extends from the coast up to 1800 m asl and crosses the locations of D17 and D47. The SMB dataset includes observations collected over the period 2004-2018 and covers

the strong SMB gradient existing between the coast and relatively drier and colder conditions inland. All annual observed values contained in each model grid cell (10 x 10 km$^2$) are averaged to produce a mean observed SMB value per grid cell.

## 5 Model evaluation

### 5.1 Near-surface climate

Accurate near-surface flows are required for a realistic representation of drifting-snow processes. The modelled annual mean

(2004-2018) 2-m wind field (Fig. 1a) shows dominant southeasterly katabatic flows all over the integration domain. Modelled annual mean wind directions at D47 (149°) and D17 (162°) agree within less than 10° with observations (Amory, 2020a). Local flow acceleration causes the highest annual mean near-surface wind speeds to be simulated in confluent topography and over 150 km along the coast east of D17 location. The occurrence of maximum wind jets along this coastal section of Adelie Land





is a well-known feature of the study area supported by modelling efforts and observations (e.g., Parish and Wendler, 1991; Wendler et al., 1993).

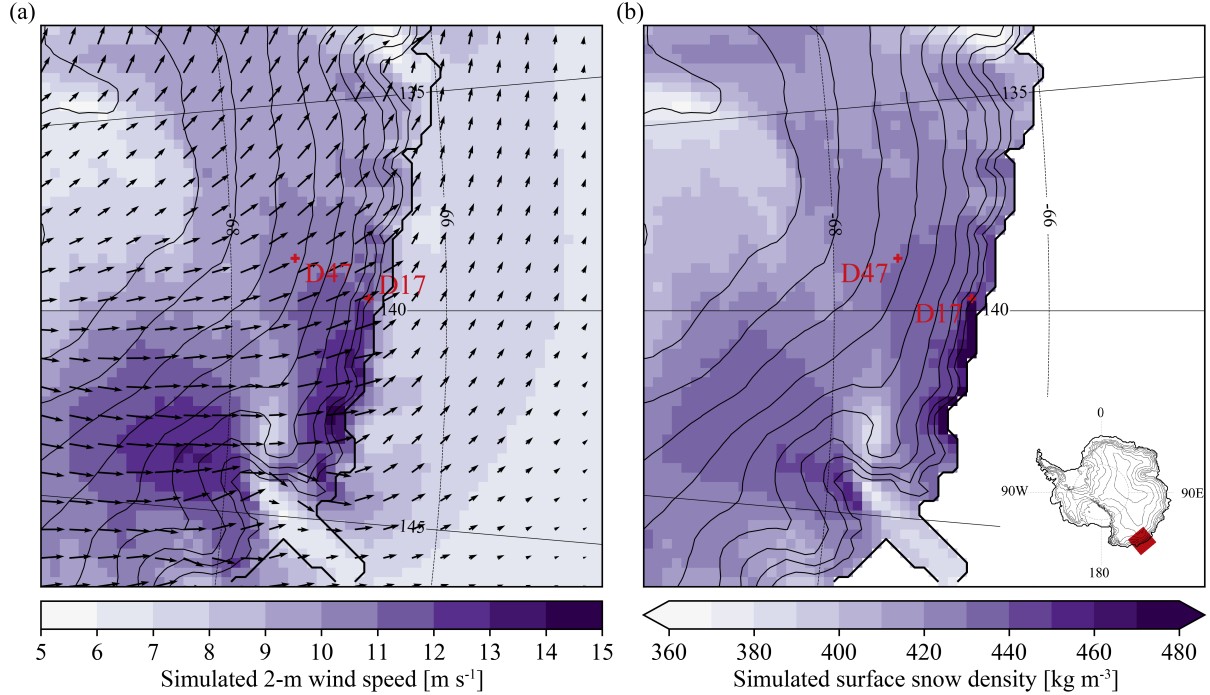

**Figure 1.** (a) Simulated annual mean (2004–2018) 2-m wind speed (colors) and direction (arrows) and (b) surface snow density on the model integration domain after removal of the relaxation zone (10 grid cells). The thin black curves show the ice-sheet topography at 250 m contour intervals. The position of the model domain on the Antarctic ice sheet is indicated in the inset. The locations of measurement sites D47 and D17 are marked with red crosses.


Snow at the surface densifies with drifting snow (Eq. 9), causing the spatial distribution of $\rho_s$ to be linked with the variability in near-surface wind speed (Fig. 1b). Lowest annual mean density values amount to 360 kg m$^{-3}$ in areas of low wind speed where melting is absent and increases in areas of flow acceleration where drifting-snow compaction is the most active. The highest $\rho_s$ values around 480 kg m$^{-3}$ are produced at the coast where both drifting-snow compaction and (summer) melting occur.


Consistent evaluation of model results for drifting-snow applications requires a fine temporal resolution since drifting snow responds to high-frequency fluctuations in wind speed. At half-hourly resolution, MAR generally underestimates 2-m wind speeds at D47, particularly above 10 m s-1, with a mean bias of -1.5 m s$^{-1}$ and a root-mean squared error (rmse) of 2.2 m s$^{-1}$ (Fig. 2a). Strong winds are better captured at D17 at the expense of weak wind conditions and model exactitude, resulting in a

positive mean bias of + 1.3 m s$^{-1}$ and a rmse of 3.2 m s$^{-1}$ (Fig. 2b). Note that these statistics, despite data quality control, might still be affected by some measurements likely subject to instrument malfunction (icing), especially at D17 due to the proximity of the ocean, when observed half-hourly wind speeds near zero are reported concurrently with considerably higher simulated



values. Actual overestimation of low wind speed values would however not be expected to be detrimental since observations show that drifting snow is usually triggered by wind speeds above 5 m s$^{-1}$ at both locations, a feature well reproduced by the model (Fig. S3). Data dispersion then reduces with increasing wind speeds.

The general underestimation in near-surface wind speed at D47 could be caused by the temperature-dependent parametrisation of $z_0$, locally still yielding too high values, while at D17 modelled $z_0$ values are closer to observations (Fig. 3). Another explanation potentially involved in the underestimation of wind speed maxima is the model misrepresentation of large eddies in case of strong winds. The local turbulence scheme of MAR is adapted for stable atmospheric boundary layers in which small eddies develop and quickly dissipate. Due to the strong turbulent mixing induced by high wind speeds, the atmospheric boundary layer is mostly statically neutral in coastal Adelie Land (Amory et al., 2017), and frequent development of drifting-snow layers of several hundreds of meter in thickness in this area (Palm et al., 2018) suggests the presence of large eddies of the height of the boundary layer. Local turbulence schemes commonly struggle to reproduce the well-mixed character of neutral atmospheric boundary layers (Hillebrandt and Kupka, 2009) more representative of coastal windy Antarctic regions, and typically fail to represent downward entrainment of momentum by large eddies from higher atmospheric levels, leading to potentially erroneous prediction of near-surface wind gusts.

The saturation vapour pressure of air is a strongly dependent function of temperature, and the relative humidity determines the potential for atmospheric sublimation. Adequate performance in modelling these fields is of prime importance for the representation of drifting-snow sublimation. Near-surface air temperature (Fig. 2c,d) is well represented at both stations (positive bias < 1 °C and centered rmse < 2 °C). As drifting-snow sublimation is a determining contributor to the atmospheric humidity budget in the area (Amory and Kittel, 2019; Le Toumelin. et al., 2020), reasonable relative humidity statistics (Fig. 2e,f) despite a remaining significant dispersion along the whole range of values suggest a realistic reproduction of this process in the lowest model levels.

The vertical resolution could limit the general ability of the model to represent the atmospheric boundary layer near the coast. Refining the vertical discretisation by doubling at the same time the total number of levels and the number of levels in the lowest 100 m however does not significantly improve model performance when evaluated against near-surface observations (See Sect. S2).

## 5.2 Drifting-snow occurrences

Since drifting snow preferentially occurs during strong wind events, general biases in the model representation of wind speed influence the representation of drifting-snow frequency (Fig. 4). Monthly frequency values are computed as the fraction of half-hourly drifting-snow occurrences with a simulated or observed near-surface drifting-snow mass flux greater than $10^{-3}$ kg m$^{-2}$ s$^{-1}$ (Amory et al., 2017). Figure 4b,c illustrates that the model reproduces the observed spatio-temporal variability in monthly drifting-snow frequency, with a correlation coefficient $r$ of 0.49 and 0.65 respectively at D47 and D17. The underestimation of strong winds at D47 leads to underestimation of drifting-snow frequency (observed and modelled averages of 0.81 and 0.61), which is then better represented by the model at D17 (observed and modelled averages of 0.61 and 0.57). Nearly consistent underestimation of drifting-snow frequency at D47 could also be caused by surface compaction that is locally too strong in the






**Figure 2.** Scatter plots of observed vs. simulated half-hourly wind speed (top), air temperature (middle) and air relative humidity with respect to ice (bottom) at 2 m height for stations D47 (left) and D17 (right). The coloured lines show the 1:1 line (dashed black) and the best linear fit (red).

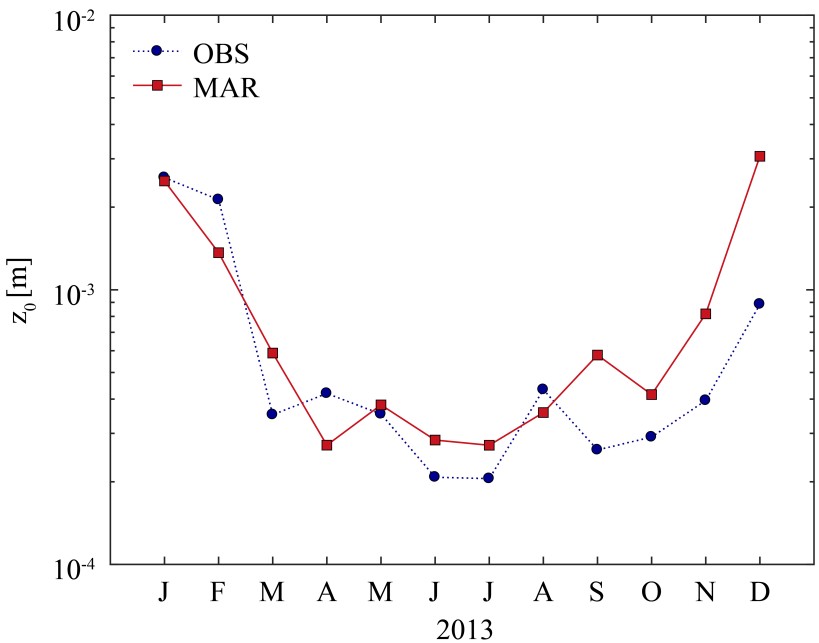

**Figure 3.** Observed (blue circles) and simulated (red squares) monthly median values of roughness length for momentum at D17 during year 2013. Observed values are taken from Amory et al. (2017)

.

model (see Sect. 6.2), shortens the duration of events and inhibits further drifting snow until snow replenishment by snowfall, while better model agreement with observations at D17 suggests a value of $\tau_{DR}$ likely better suited for this location. Note that modelled drifting snow is not the most frequent where the strongest wind speeds are found (Fig. 4a). Highest frequency values

are simulated east of D17 and D47 over topographic crests upstream of the Mertz Glacier (67.5° S, 144.8° E) where orographic lifting produces enhanced precipitation rates which then contribute to favor the occurrence of drifting snow by i) increasing the airborne snow mass independently from erosion and ii) decreasing $\rho_s$ and lowering the erosion threshold.

Figure 4 only compares monthly frequency values. Typical durations of drifting-snow events in Adelie Land, however, range from several h to a few days at most (Amory, 2020a). To evaluate the model results at higher temporal resolution closer to

characteristic time scales of drifting snow, half-hourly data are used to compute the probability of detection ($POD$) and false alarm ratio ($FAR$)

$$POD = 100\frac{a}{a+b}; FAR = 100\frac{c}{c+a} \tag{12}$$

where a is the number of half-hourly drifting-snow occurrences correctly simulated, b the number of occurrences missed by the model and c the number of occurrences simulated but not observed. The Rousseau index ($RI$), a measure of model

predictability originally defined to assess rainfall forecasts (Rousseau, 1980) and already used to evaluate modelled drifting-



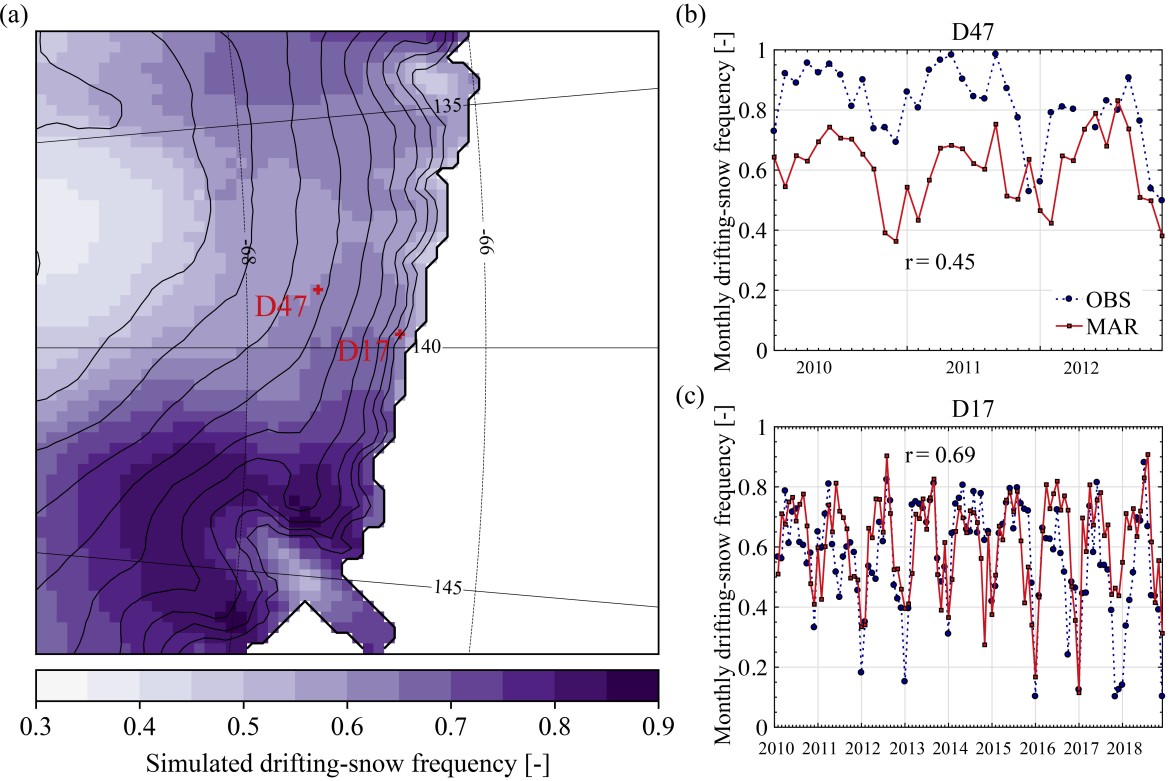

**Figure 4.** (a) Simulated annual mean (2004–2018) drifting-snow frequency. (b) Monthly drifting-snow frequency from observed (circles, blue dashed curve) and simulated (squares, red solid curve) half-hourly drifting-snow mass fluxes at D47 and (c) D17. Monthly frequency values are calculated as the fraction of half-hourly drifting-snow occurrences (near-surface drifting-snow mass fluxes greater than $10^{-3}$ kg m$^{-2}$ s$^{-1}$ in both the model and observations.

snow occurrence in an alpine context (Vionnet et al., 2013), is also calculated. $RI$ varies between -100 and 100. A negative value means that the model is less successful than a estimation entirely based on climatology, 0 indicates no skill, and 100 is obtained for a perfect simulation (see Appendix A for mathematical details).

Positive $RI$ values are obtained at both locations (Table 2), meaning the model ability to predict drifting-snow occurrences at the monthly scale (Fig. 4b,c) arises from a reasonable reproduction of their actual timing and duration. MAR shows better results (higher $POD$ and $RI$) at D17 than at D47, but also simulates more unobserved occurrences (higher $FAR$) that compensate for missed occurrences when evaluating model results at coarser temporal resolution.

## 5.3 Drifting-snow transport

Qualitative evaluation of MAR against monthly cumulative near-surface drifting-snow mass fluxes at D47 and D17 (Fig. 5b,c)
reveals that the model captures the general temporal evolution of drifting-snow transport at both locations ($r = 0.64$ and 0.89). As a result of interactions between modelled near-surface flow conditions and surface snow properties, drifting snow

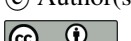



**Table 2.** Statistical evaluation of the ability of MAR to simulate half-hourly drifting-snow occurrences at D47 (2010–2012) and D17 (2010–2018). See Sect. 5.2 for the definition of $POD$, $FAR$ and $RI$.

| Station | $POD$ | $FAR$ | $RI$ |
|---------|-------|-------|------|
| D47 | 64.5 | 13.4 | 9.1 |
| D17 | 80.9 | 25.4 | 45.5 |

**Table 3.** Observed and simulated total drifting-snow transport ($10^6$ kg m$^{-2}$) from 0 to 2 metres above surface at D47 and D17. A distinction is made between total snow transport computed from observed (OBS) and simulated (MAR$_{obs}$) near-surface drifting-snow mass fluxes cumulated over each drifting-snow event identified in the database and over each simulated drifting-snow event (MAR$_{sim}$).

| Station | OBS | MAR$_{obs}$ | MAR$_{sim}$ |
|---------|-----|-------------|-------------|
| D47 | 1.51 | 1.84 | 1.88 |
| D17 | 5.25 | 4.92 | 5.4 |

is subject to a high variability in space and time. This variability is simulated by the model with alternating underestimation and overestimation of drifting-snow transport depending on the time period and location. Figure 6 compares observed and simulated snow transport during each observed drifting-snow event. As in Amory (2020a), a drifting-snow event is defined as a period over which the observed snow mass flux is above the detection threshold of $10^{-3}$ kg m$^{-2}$ s$^{-1}$ for a minimum duration of 4 h. Discrepancy between model and observations is larger for (shorter) events of lower magnitude ($< 10^3$ kg m$^{-2}$ s$^{-1}$) and reduces for longer, more important events that contribute predominantly to the local drifting-snow transport (Amory, 2020a).

Table 3 compares the total snow transport during observed drifting-snow events (OBS) to that estimated by the model during these observed events (MAR$_{obs}$) and during every simulated events (MAR$_{sim}$). Referring only to observed events, MAR respectively overestimates (+21.6 %) and underestimates (-6.1 %) drifting-snow transport at D47 and D17. Together with the consistent underestimation of drifting-snow frequency at D47 (Fig. 4b), this means that the model simulates the main events but underestimates the occurrence of events of lower magnitude and associated transport at this location (Fig. 6). However, overestimation of drifting-snow transport by the model during observed events overcompensates for the missed events, with a slight contribution of false alarms to the simulated total drifting-snow transport (MAR$_{sim} \sim$ MAR$_{obs} >$ OBS). At D17, while the simulated drifting-snow frequency (Fig. 4c) and total drifting-snow transport during observed events are in closer agreement with the observations, simulated but unobserved drifting snow is more significant (Table 3) and result in overestimation of total drifting-snow transport (MAR$_{sim} >$ MAR$_{obs} \sim$ OBS).

Analysing the relationship between modelled wind speed and drifting-snow transport demonstrates that the model performance varies according to the flow conditions leading to drifting snow. Median relative biases in mean 2-m wind speed and cumulative near-surface transport during drifting-snow events are shown at both measurement sites for 2-m wind-speed bins of 1 m s$^{-1}$ in Fig. 7 for the most represented wind-speed categories (8 to 17 m s$^{-1}$). Since wind speed and drifting-snow transport



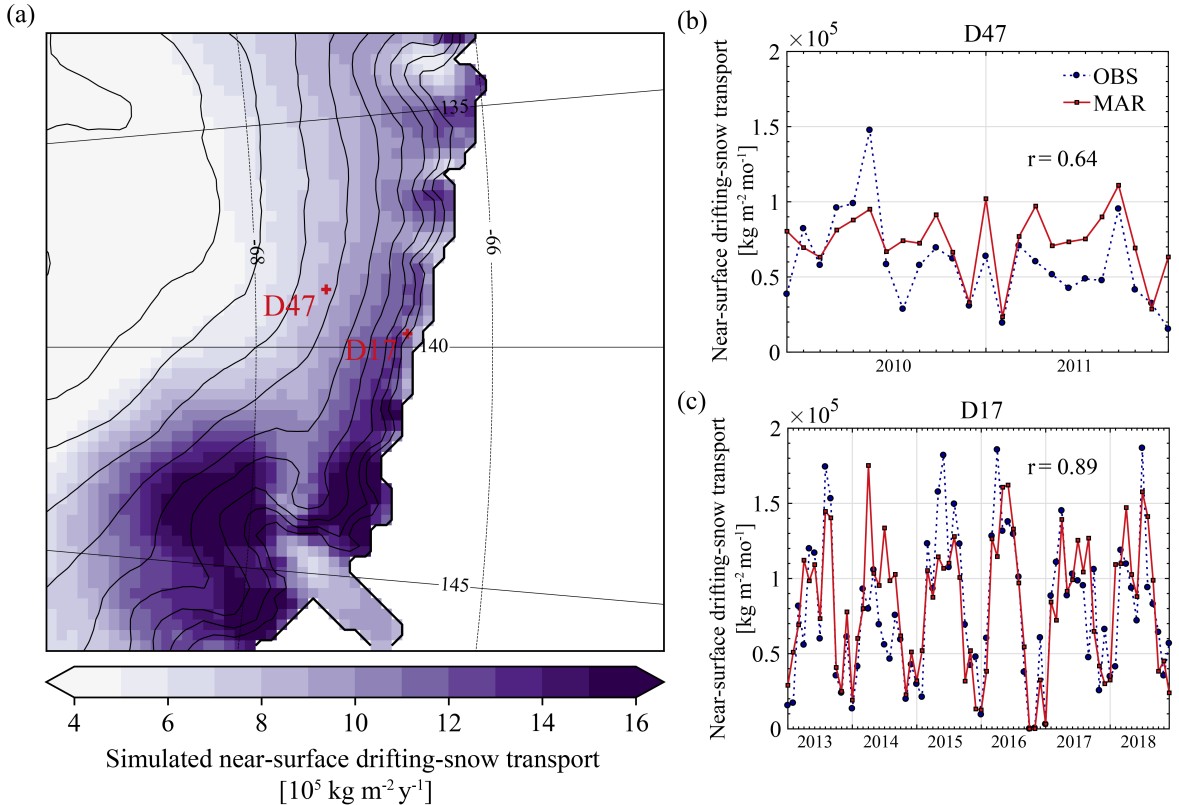

**Figure 5.** (a) Simulated annual mean (2004–2018) near-surface drifting-snow transport. (b) Monthly near-surface drifting-snow transport from observed (circles, blue dashed curve) and simulated (squares, red solid curve) half-hourly drifting-snow mass fluxes at D47 and (c) D17.

are monotonically related, biases in event-averaged wind speed can explain biases in cumulative drifting-snow transport when both are of the same sign. Figure 7 shows fluctuations in the sign of the bias in drifting-snow transport when biases in mean wind speed remain of constant sign. This suggests that model errors in terms of drifting-snow transport can also be attributed

to the erosion, microphysical and/or turbulence schemes. Nevertheless, note that uncertainties in the observations can also affect the evaluation. In particular, while 2G-FlowCapt™ sensors can detect the occurrence of snow transport with a high level of confidence (Trouvilliez et al., 2015), its ability to estimate drifting-snow mass fluxes remains to be assessed in Antarctic conditions (Amory, 2020a).

The mass transported in drifting snow has been shown to correlate with wind speed in a power-law fashion (e.g., Budd,

1966; Radok, 1977; Mann et al., 2000; Amory, 2020a). Figure 8, in which the modelled annual mean of horizontal near-surface drifting-snow transport is studied as a function of 2-m wind speed for all continental grid cells, shows that this relationship is well reproduced by the model. This enables to assess the plausibility of the model results in the absence of observations in other locations of the integration domain, and explains the spatial distribution of drifting-snow transport, with maximum



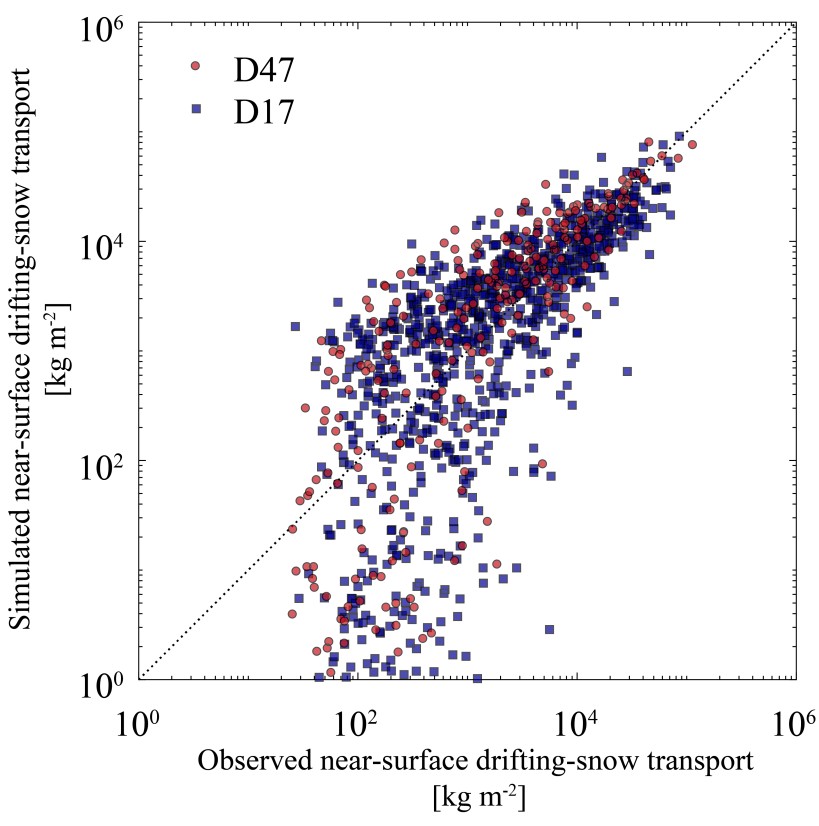

h

**Figure 6.** Scatter plots of observed vs. simulated near-surface drifting-snow transport for each drifting-snow event at D47 (red circles) and D17 (blue squares). As in Amory (2020a), a drifting-snow event is defined as a period over which the observed snow mass flux is above the detection threshold of $10^{-3}$ kg m$^{-2}$ s$^{-1}$ for a minimum duration of 4 h.

values generally simulated in areas of highest 2-m wind speeds (Fig. 5a). The dispersion around a given wind speed value (Fig.

8) results from variations in the erosion threshold (Mann et al., 2000; Amory, 2020a). Departure from the power-law relation at low wind speeds is caused by a predominant contribution of snowfall to the airborne snow mass (Amory et al., 2015) and corresponds to areas of near-surface flow deceleration, such as over the Mertz Glacier ice tongue or in topographic depressions.

## 5.4 Surface mass balance

In the current version of MAR, precipitating, eroded, and deposited snow particles are all included in the snow particles ratio $q_s$. Precipitation, erosion and deposition can occur simultaneously and repeatedly within the same grid cell. Simulated snowfall ($SF$) and erosion ($ER$) amounts then respectively account for the cumulative snow mass that is transferred to and

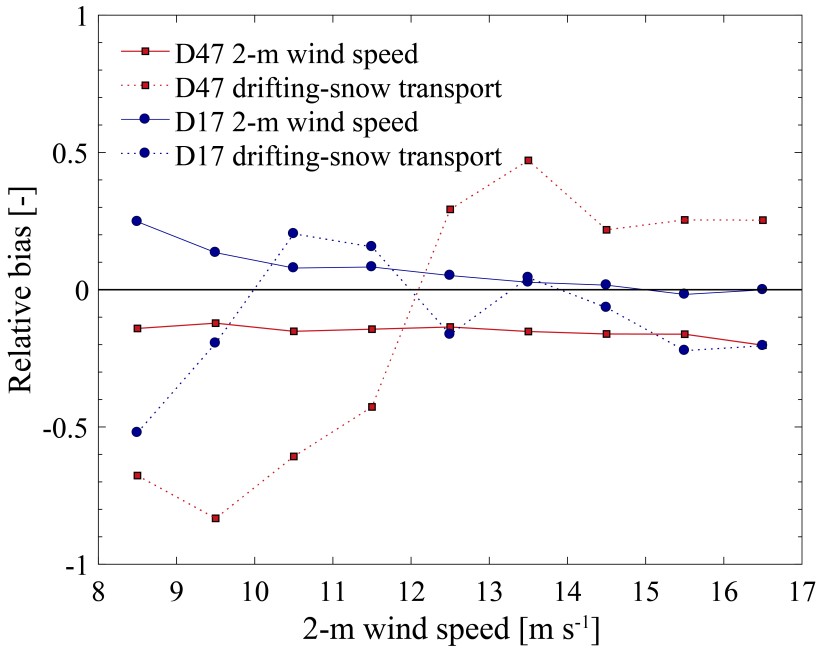

**Figure 7.** Relative bias in mean wind-speed (solid curves) and cumulated snow transport (dashed curves) during drifting-snow events. Relative biases are expressed as the ratio of the difference between model and observation to the observation and shown as median values within wind-speed bins of 1 m s$^{-1}$ at D47 (red curves) and D17 (blue curves).

removed from the surface. Each of the two components thus cannot be used individually to determine the integrative contribution of snowfall, erosion and deposition separately when the drifting-snow scheme is switched on. From that perspective an

erosion/deposition index $i_{E/D}$ is defined as

$$i_{E/D} = \frac{SF}{ER} \qquad (13)$$

which reflects the relative local proportion of eroded to deposited snow mass.

Figure 9 compares the simulated and observed mean annual SMB and shows $i_{E/D}$ along the stake transect whose location is marked with black dots on Fig. 10. MAR represents the general variability in SMB with a strong increase over the first

tens of kilometers from the coast. The mean SMB bias is negative with ∼67 kg kg m$^{-2}$ y$^{-1}$ (20 %) but is mainly due to underestimation of the simulated SMB near the coast. This results from either overestimation of erosion or underestimation of precipitation, or a combination of both, leading to the highest values of $i_{E/D}$ along the transect. The lowest, negative simulated SMB value is not found at the coast but a few grid cells inland where $i_{E/D}$ is maximal, i.e. where erosion amounts to 93 % of the snow mass accumulated locally. Simulated surface sublimation (and negligibly runoff) also contributes but to a lesser

extent to net ablation at the surface (not shown). More generally, ablation is simulated where $i_{E/D}$ displays maximum values



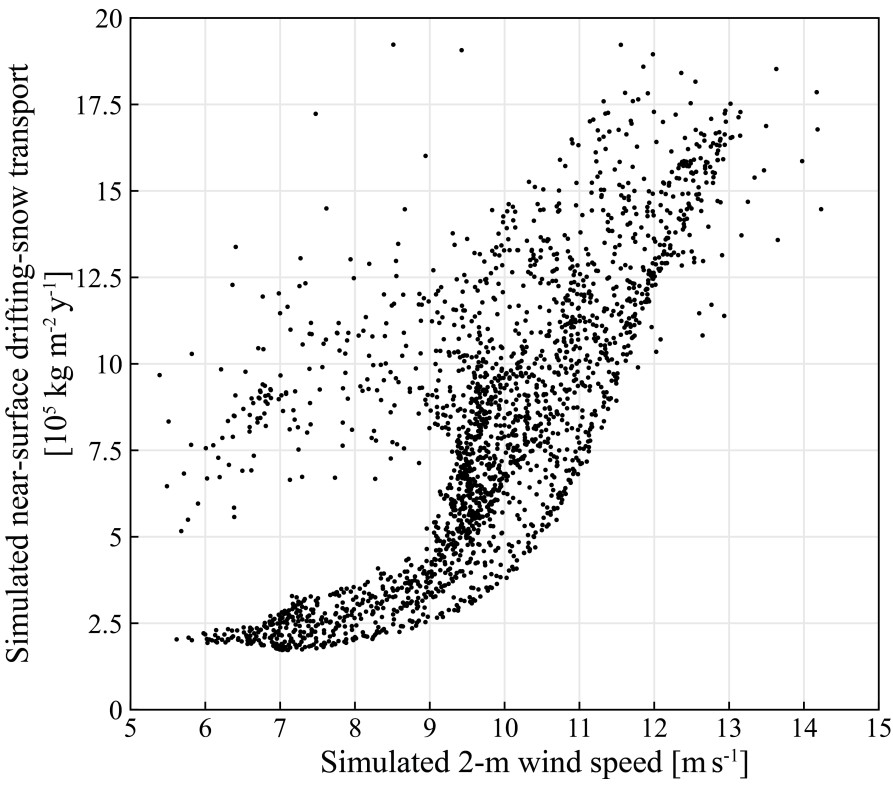

**Figure 8.** Simulated mean annual (2004–2018) near-surface drifting-snow transport as a function of 2-m wind speed at all continental grid cells.

(Fig. 10). This is in accordance with the literature which suggests that drifting-snow processes are the main driver behind the formation of blue-ice areas in Adelie Land (Genthon et al., 2007; Favier et al., 2011).

Although modelled and observed SMB values are in relative good agreement, the raw observed SMB signal (i.e., not spatially averaged over model grid cells) highlights an important subgrid variability in SMB gradients that is still not resolved at 10 km resolution (Fig. 9). It also shows that the modelled SMB decrease a few kilometers upstream of the coastline is also retrieved in the observations but occurs more locally. This suggests that an improved representation of the spatial variability in SMB in the area with the model would require an even higher resolution to better resolve the interactions between the atmosphere and surface topography.

Model limitations also relate to the size of the integration domain since the model is not driven at its lateral boundaries by drifting-snow fluxes. This might affect the results, notably the location of ablation areas close to or directly downwind of the model boundaries where drifting snow can only diverge in the absence of possible advection from upwind areas outside of the integration domain.



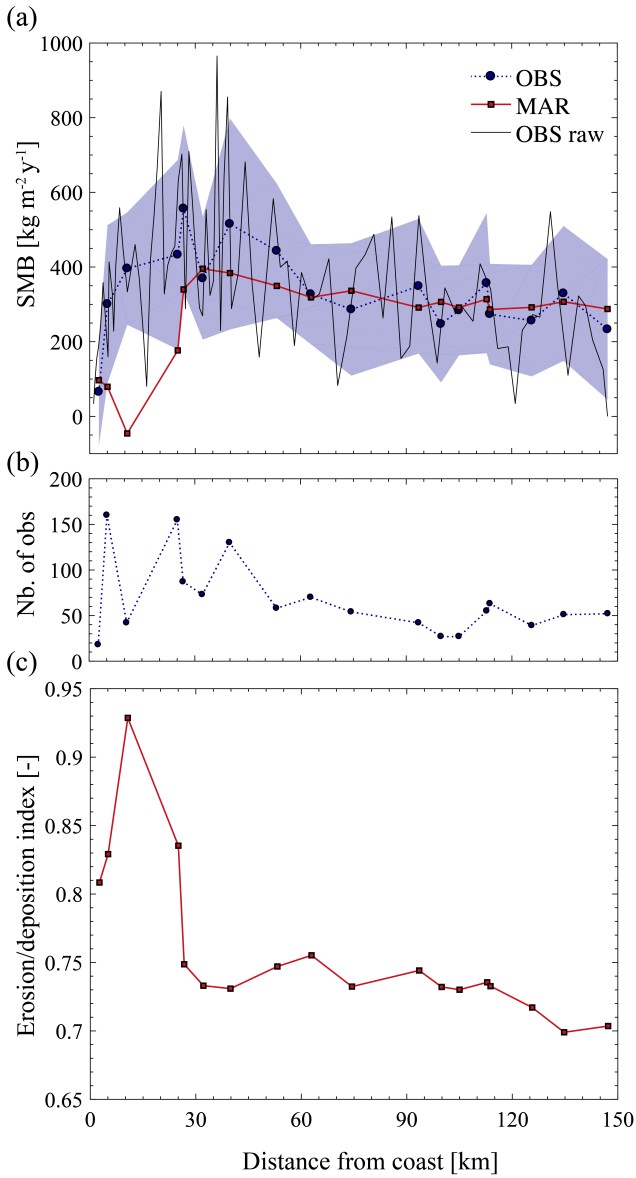

**Figure 9.** (a) Simulated (squares, red solid curve) vs. observed (circles, blue dotted curve) mean annual SMB (2004–2018) along the stake transect. The location of the SMB observations is marked with black dots in Fig. 10. Observed annual SMB values are averaged on MAR grid cells with no interpolation nor weighting. The spatial resolution is 10 km. Distance along the transect starts at the coast, and is computed as the average distance of all annual observations contained in each grid cell. Uncertainty of observed SMB (blue shaded area) is approximated by the standard deviation of all annual observations contained in each grid cell and is shown together with the mean annual (2004-2018) raw SMB signal along the transect (thin black curve) as an illustration of subgrid variability. (b) Number of cumulated annual observations per grid cell along the transect. (c) Erosion/deposition index along the transect.



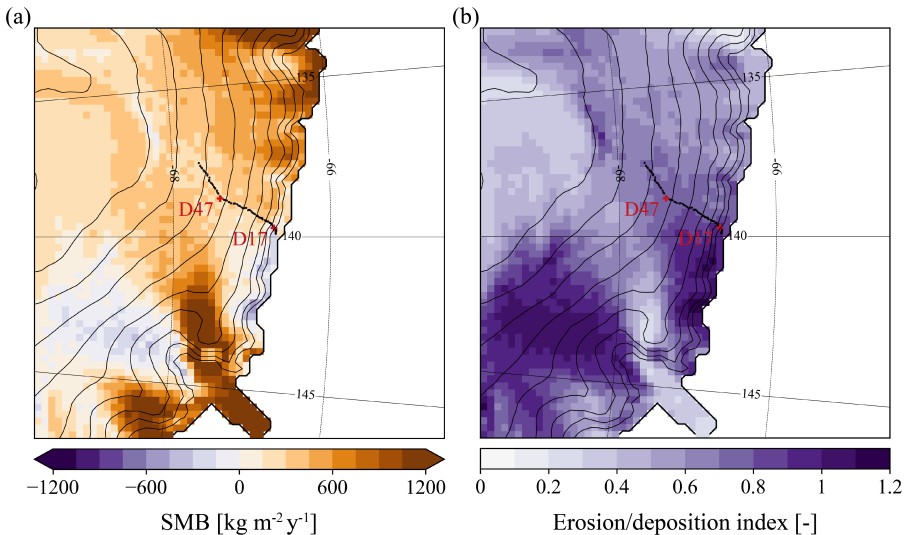

**Figure 10.** (a) Simulated mean annual (2004–2018) SMB and (b) corresponding erosion/deposition index.

## 6 Sensitivity analysis

### 6.1 Sensitivity to the model version

Improvements in the simulation of drifting snow with the newest version of MAR is illustrated in Fig. 11 for the month of January 2011 at site D47, as studied in Amory et al. (2015). Despite a coarser horizontal resolution (10 km vs 5 km) and knowing that different reanalysis (ERA5 vs ERA-Interim) were used as forcing, MAR displays enhanced capabilities in simulating the timing, duration and magnitude of drifting snow (Fig. 11a) relative to the previous model version and setup (MARv2) used in Amory et al. (2015). Since the representation of 2-m wind speed only slightly improves with the current

version (not shown), the better agreement with observations can be mainly attributed to a higher performance of the drifting-snow scheme. Improved drifting snow in MAR also leads to increased 2-m relative humidity as a result of enhanced atmospheric sublimation (Fig. 11b). Periods of near-saturated conditions, whose occurrence corresponds with that of drifting snow, are better captured in MAR than in the former model version.

### 6.2 Sensitivity to input parameters

The performance of MAR is dependent on the choices made for several input parameters of the drifting-snow scheme. Sensitivity experiments were undertaken to assess the sensitivity of the drifting-snow scheme to changes in the fresh snow density $\rho_0$ and characteristic time scale for drifting-snow compaction $\tau_{DR}$ over a range of otherwise plausible values. Sensitivity to the neglection of drifting-snow compaction and the parameterisation of the threshold friction velocity for snow erosion $u_{*t}$ was also explored. Simulated drifting-snow frequency and transport is evaluated for each experiment at site D47 for year 2010 and

D17 for year 2013. Results are summarised in Table 4.



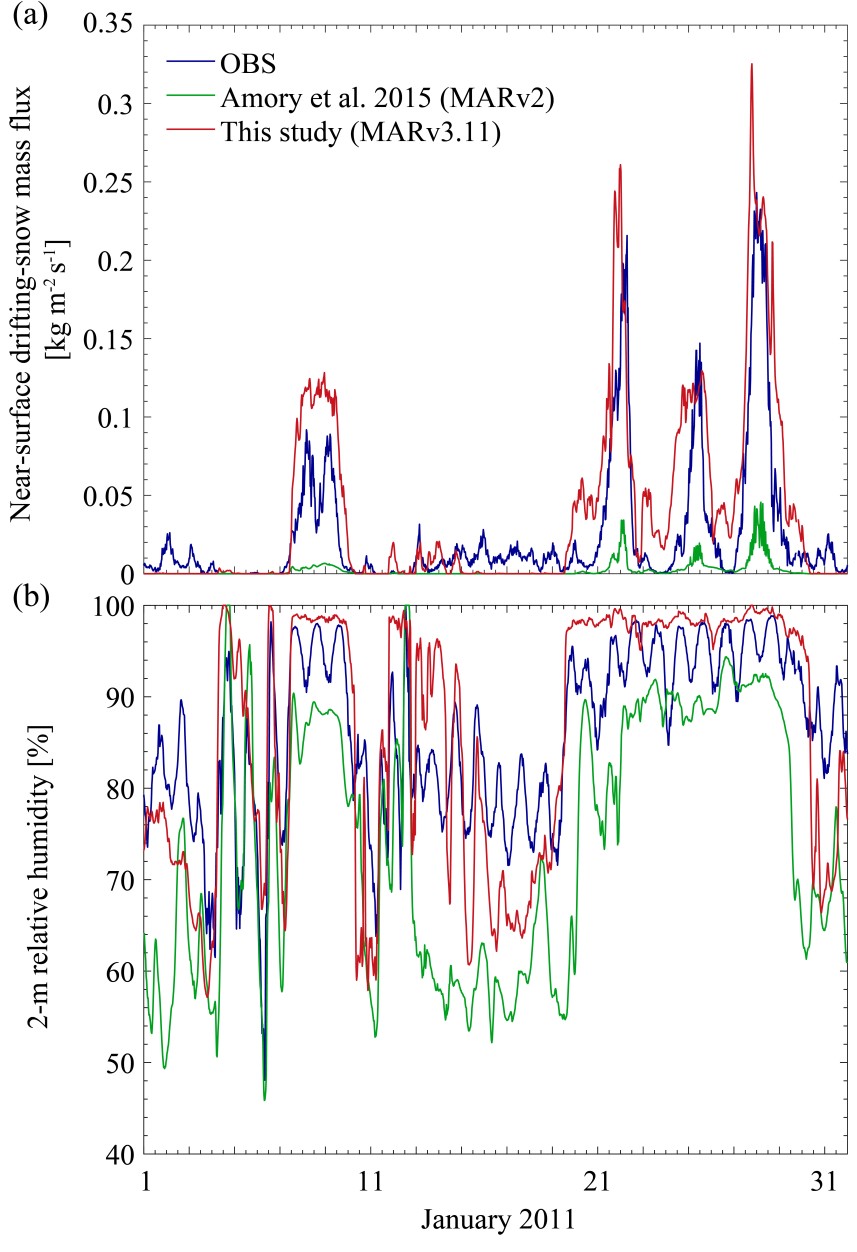

**Figure 11.** (a) Near-surface drifting-snow mass flux and (b) 2-m relative humidity in MARv2 (green) and MARv3.11 (red) vs. observations (blue) for the case study in January 2011 at D47 discussed in Amory et al. (2015). Note that MARv3.11 is run at 10 km resolution and forced by ERA5 while MARv2 is run at 5 km and forced by ERA-Interim.





**Table 4.** Sensitivity analysis to various model parameters of the annual drifting-snow frequency and accumulated drifting-snow transport up to 2 m above the surface in $10^5$ kg m$^{-2}$ y$^{-1}$ during year 2010 at D47 and 2013 at D17. Values in brackets refer to the relative difference in % compared to observations. L07 and V12 denotes adaptation of the parameterisation of $u_{*t}$ according to Liston et al. (2007) and Vionnet et al. (2012) respectively (see Appendix B).

| Experiment | D47 | | D17 | |
|---|---|---|---|---|
| | Frequency | Snow transport | Frequency | Snow transport |
| 2G-FlowCapt™ | 0.84 | 8.28 | 0.61 | 9.15 |
| Control: $\rho_0 = 300$ kg m$^{-3}$, $\tau_{DR} = 24$ h | 0.63 | 8.81 (+6.4 %) | 0.60 | 9.71 (+6.1 %) |
| $\rho_0 = 250$ kg m$^{-3}$ | 0.84 | 13 (+57 %) | 0.34 | 4.54 (-50.4 %) |
| $\rho_0 = 350$ kg m$^{-3}$ | 0.91 | 15.4 (+86 %) | 0.89 | 15.1 (+65 %) |
| $\tau_{DR} = 12$ h | 0.59 | 7.9 (-4.6 %) | 0.55 | 8.88 (-3 %) |
| $\tau_{DR} = 48$ h | 0.72 | 10.81 (+30.6 %) | 0.68 | 11.37 (+24.3 %) |
| $\rho_{DR} = 300$ kg m$^{-3}$, $\tau_{DR} = \infty$ | 0.98 | 17.52 (+111.6 %) | 0.97 | 16.47 (+80 %) |
| $u_{*t} = $ L07 | 0.46 | 4.71 (-43.1 %) | 0.32 | 3.94 (-56.9 %) |
| $u_{*t} = $ V12 | 0.46 | 4.5 (-45.6 %) | 0.31 | 3.95 (-56.8 %) |

The model exhibits a high sensitivity to $\rho_0$. This parameter determines the density at which fresh snow is deposited but also intervenes in the definition of $u_{*t}$ (Eq. (1)) and drifting-snow compaction rate (Eq. (9)). To conserve the physical consistency of the drifting-snow scheme, changes in $\rho_0$ must be applied to the whole set of parametrisations and necessarily leads to modifications that result from the combined sensitivities to each of the three parameterisations. Contrasting results between D47 and D17 demonstrate that the model response can be spatially heterogeneous depending on dominant local effects. Reducing (increasing) $\rho_0$ to 250 (350) kg m$^{-3}$ at D17 leads to lower (higher) snow density values at deposition by snowfall but also increases (decreases) $u_{*t}$ (Fig. 12), resulting in a general dampening (enhancement) of simulated drifting snow around 50 % in both frequency and transport. This indicates that lower surface snow density $\rho_s$ is locally overcompensated by a corresponding strong increase in $u_{*t}$. The model would become less and less sensitive to $\rho_0$ values below 250 kg m$^{-3}$ because $u_{*t}$ would more rapidly reach high prohibitive values for snow erosion. The reverse situation is depicted at D47, where the model simulates lower $\rho_s$ (Fig. 1b). Lowering $\rho_0$ to 250 kg m$^{-3}$ induces a local decrease in $\rho_s$ that improves drifting-snow frequency but produces drifting-snow transport beyond observed values ($> 50$ %) despite higher $u_{*t}$. Overestimated and similar drifting-snow features are obtained at both locations with $\rho_0 = 350$ kg m$^{-3}$. Discussing the influence of $\rho_0$ beyond 350 kg m$^{-3}$ would implicitly imply deposition of fresh snow in a well-advanced state of compaction and is therefore not considered

Changes in $\tau_{DR}$ alter the timing and duration of drifting-snow events but appear to exert a moderate influence on modelled drifting snow. Halving and doubling $\tau_{DR}$ to 12 h and 48 h respectively decreases and increases local drifting-snow frequency and transport with a relatively lower magnitude ($< 30$ %) than the tested changes in $\rho_0$. The value of $\tau_{DR} = 24$ h is of local





significance since it has been scaled on the observed median duration of drifting-snow events in Adelie Land. Different time
scales for drifting-snow compaction could be expected under different environmental conditions, particularly for different wind
speed regimes. This would suggest that $\tau_{DR}$ should be varying regionally, and if needed, could be adapted to more practical
values depending on the local climate of the region of interest.

Disabling drifting-snow compaction by setting $\tau_{DR}$ to infinity and the density of deposited drifting snow $\rho_{DR}$ to the standard
$\rho_0$ value of 300 kg m$^{-3}$ produces almost permanent drifting snow and leads to the strongest overestimation ($> 100$ % at D47)
of drifting-snow transport among the sensitivity experiments. These model results illustrate the necessity of taking the negative
feedback of drifting-snow compaction into account to simulate drifting-snow frequency and mass fluxes satisfactorily at the
two measurement sites and prevent removal of almost all the accumulated snow.

Conversely, the strongest inhibition of drifting snow (by 50 %) is obtained through changes in $u_{*t}$ following the parame-
terisations of Liston et al. (2007) and Vionnet et al. (2012), hereafter referred to as L07 and V12. Appendix B gives numerical
details. The resemblance in the evolution of $u_{*t}$ with $\rho_s$ (Fig. 12) explains the similar model results obtained at both stations
with L07 and V12. However, both parameterisations do not depend on the same set of variables. L07 only depends on $\rho_s$ and
enables investigation of the influence of $\rho_0$ without inherently altering $u_{*t}$ nor $\tau_{DR}$. Overall, L07 is a stricter criterion for snow
erosion than the control parameterisation to the extent that it cannot be offset by prescribing low $\rho_0$ values nor by increasing
$\tau_{DR}$ within the range of tested values (not shown). V12 corresponds to a weighting of the erodibility index $i_{ER}$ by $\rho_s$ and
remains dependent on $\rho_0$. Like L07, V12 yields a general increase in $u_{*t}$ that becomes very restrictive and that a change in
$\tau_{DR}$ can hardly compensate for without increasing it substantially nor by increasing $\rho_0$ (not shown). The resulting inhibition
at D17 is also quite close to that obtained with $\rho_0 = 250$ kg m$^{-3}$, where $\rho_s$ is high enough for $u_{*t}$ to drive the model response
to changes in $\rho_0$.

Table 4 shows that the best agreement between model and observations is obtained with the control setup. However, this
agreement also most likely results from error compensations which are difficult to quantify and identify in the model. Al-
though model results remain in the same order of magnitude as observations in most sensitivity experiments, the simulated
drifting-snow frequency and transport appear very sensitive to input parameters, in particular to the choice of $\rho_0$ and the pa-
rameterisation of $u_{*t}$ with the control setup, which inherently also affect the SMB. It is likely that different combinations of
these parameters could lead to equally satisfactory model results, providing that the SMB is also conjointly assessed to preserve
consistency between erosion and accumulation rates.

**7 Conclusion**

A dataset of drifting-snow and SMB observations collected in Adelie Land has been used to evaluate the latest version of the
regional climate model MAR (v3.11) equipped with a new drifting-snow scheme. Modifications and developments relative
to the previous drifting-snow scheme mainly consisted in i) computing the roughness length $z_0$ as a varying function of air
temperature and disabling the drag-partition scheme and its inhibiting effect on snow erosion, ii) rewriting the parameterisation
of the threshold friction velocity for snow erosion $u_{*t}$ as a function of surface snow density only, iii) differentiating snow density

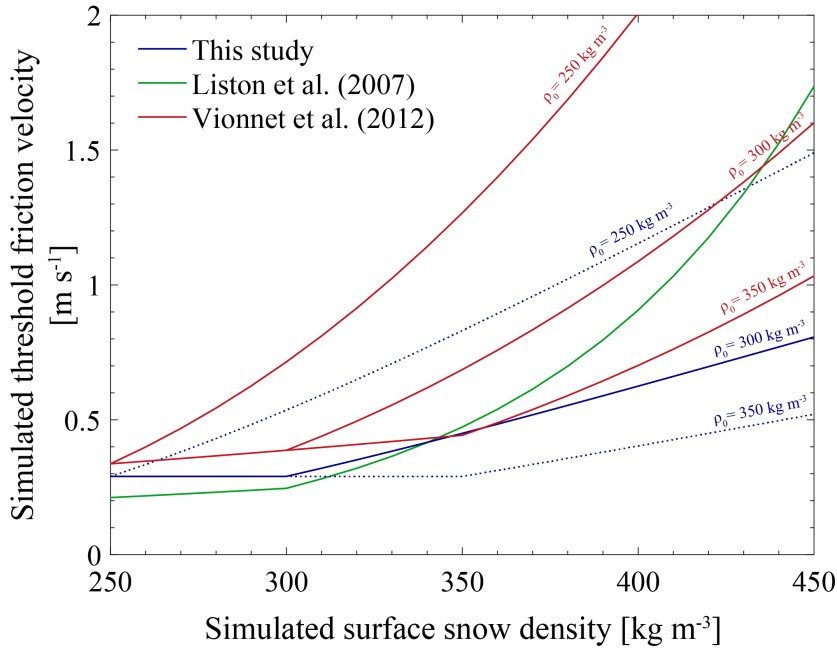

**Figure 12.** Threshold friction velocity ($u_{*t}$) as a function of surface snow density ($\rho_s$) for three sets of parameterisations and different fresh snow density values ($\rho_0$). A constant roughness length for momentum $z_0$ of $10^{-3}$ m is assumed here for the computation of $u_{*t}$ (see Sect. 3). The solid blue line highlights the current control parameterisation of $u_{*t}$ in MAR ($\rho_0 = 300$ kg m$^{-3}$).

at deposition between precipitation and drifting snow, iv) including a simple parameterisation for drifting-snow compaction, and v) restricting erosion within one model time step to the uppermost snowpack layer only. The model provides improved results compared to a previous version and demonstrates a good capability to reproduce drifting-snow occurrences and near-surface transport up to the scale of the drifting-snow event at two measurement sites 100 km apart over multiple-year periods,

conjointly with the resulting variability in SMB along a 150 km long transect crossing the location of the measurement sites. These results constitute an encouraging step toward the use of MAR to study drifting snow and its influence on the Antarctic climate and SMB over a wide range of spatial and temporal scales from local case studies to a continent-wide climatology.

Modelling near-surface drifting snow could be expected to require a particularly fine vertical resolution close to the surface to resolve the mass exchange with the saltation layer and the strongly decreasing snow mass fluxes in the first metres above

ground. A lowest atmospheric level at 2 m however does not appear as a limiting factor compared to observed snow mass fluxes vertically integrated from 0 to 2 m above ground, suggesting that accounting for drifting-snow physics as proposed in MAR could be envisaged in global climate modelling. Future developments should focus on the distinction between snow particles remobilised from the ground from precipitating particles to explicitly account for different particle properties such as snow grain shape and size or settling velocity, and enable disentanglement of the contribution of snowfall from eroded particles to

the local climate and drifting-snow features.



Model evaluation in this study is limited to the surface (< 2 m). Snow-mass transport and atmospheric sublimation processes during drifting snow are for a large part supplied with particles originating from the surface in the lower atmosphere, so a good representation of drifting-snow features near the surface is of prime importance. The performance of the model at higher atmospheric levels where drifting-snow transport and sublimation can supposedly be of substantial significance in terms of
SMB (e.g., Amory and Kittel, 2019; Le Toumelin. et al., 2020) is however not known yet and remains to be assessed. Remotely sensed properties of drifting snow from space (layer depth, optical thickness) recently made available through the ICESat-2 mission (Markus et al., 2017) at an unprecedented level of spatial and temporal resolution would be a great candidate for such an exercise.

*Code and data availability.* The meteorological and drifting-snow data (Amory et al., 2020) can be downloaded at https://zenodo.org/record/3630497
(last access: 30 November 2020). The surface mass balance data are available at pp.ige-grenoble.fr/pageperso/faviervi/stakeline.php. This service is provided by the National Observation Service GLACIOCLIM (https://glacioclim.osug.fr/?lang=en) piloted by the OSUG (Observatories of Earth Sciences and Astronomy of Grenoble, France). The National Observation Service GLACIOCLIM have requested that we make the data available through the dedicated webpage hosted by the OSUG website. However, we have also archived a specific version of the dataset used in this study together with the MAR source code and outputs for replication of this study at https://zenodo.org/record/4314872
(Amory, 2020b). The MAR source code is tagged as v3.11.2 on https://gitlab.com/Mar-Group/MARv3.7. See http://www.mar. cnrs.fr for more information about downloading MAR (last access: 10 December 2020).

.

# Appendix A: Rousseau index

The Rousseau index $RI$ has been originally defined to assess rainfall forecast (Rousseau, 1980). $RI$ takes into account the
number of half-hourly drifting-snow occurrences correctly simulated a, the number of occurrences missed by the model b, the number of occurrences simulated but not observed c, and the number of occurrences for which the absence of drifting-snow is correctly simulated, d, such as

$$RI = 100\frac{ad - \frac{(b+c)^2}{2}}{(a + \frac{b+c}{2})(d + \frac{b+c}{2})} \tag{A1}$$





## Appendix B: Additional parameterizations of the threshold friction velocity

Changes in the threshold friction velocity for snow erosion $u_{*t}$ was done following the parameterisations of Liston et al. (2007) and Vionnet et al. (2012). In Liston et al. (2007), $u_{*t}$ is parameterised a function of surface snow density $\rho_s$ only

$$u_{*t} = \begin{cases} 0.005 exp(0.013\rho_s) & 300 < \rho_s \leq 450 \\ 0.1 exp(0.003\rho_s) & \rho_s \leq 300 \end{cases} \tag{B1}$$

In Vionnet et al. (2012) the definition of the erodibility index $i_{ER}$ is completed with a term depending on the surface snow density $F(\rho_s)$ to account for high polar surface snow densities (Libois et al., 2014)

$$i_{ER} = 0.34 i_{ER} + 0.66 F(\rho_s) \tag{B2}$$

$$F(\rho_s) = 1.25 - 0.0042(\rho_s - 50) \tag{B3}$$

Both parameterisations are given for $\rho_s$ = 450 kg m$^{-3}$ in agreement with the maximum snow density that can be attained through drifting-snow compaction in MAR.

*Author contributions.* CAm, CK and XF designed the study. CAm and VF collected data on the field. CAm implemented model developments and updates with the help of CK and XF. CAm ran the simulations, performed the analysis and wrote the first draft of the manuscript. All authors discussed and revised the manuscript.

*Competing interests.* The authors declare that they have no conflict of interests.

*Acknowledgements.* This work would not have been possible without the financial and logistical support of the French Polar Institute
IPEV (program CALVA-1013 and GLACIOCLIM-SAMBA-411). The authors thank all the on-site personnel in Dumont d'Urville and Cap Prud'homme for their precious help in the field, in particular Philippe Dordhain for electronic and technical support. Luc Piard and Christophe Genthon are also acknowledged for their investment in collecting data and maintaining the observation system in Adelie Land, as well as Hubert Gallée for fruitful discussions. Computational resources have been provided by the Consortium des Équipements de Calcul Intensif (CÉCI), funded by the Fonds de la Recherche Scientifique de Belgique (F.R.S. – FNRS) under grant no. 2.5020.11, and the Tier-
1 supercomputer (Zenobe) of the Fédération Wallonie-Bruxelles infrastructure funded by the Walloon Region under grant agreement no. 1117545. Charles Amory is a postdoctoral researcher from the Fonds de la Recherche Scientifique de Belgique (F.R.S.-FNRS).



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
