# Peer review of "Performance of MAR (v3.11) in simulating the drifting-snow climate and surface mass balance of Adelie Land, East Antarctica"

_Geoscientific Model Development, 2020_

## Short Comment (SC1) · 21 Dec 2020

Dear authors,

in my role as Executive editor of GMD, I would like to bring to your attention our Editorial version 1.2:

https://www.geosci-model-dev.net/12/2215/2019/

This highlights some requirements of papers published in GMD, which is also available on the GMD website in the 'Manuscript Types' section: http://www.geoscientific-model-development.net/submission/manuscript_types.html

In particular, please note that for your paper, the following requirement has not been met in the Discussions paper:

- Code must be published on a persistent public archive with a unique identifier for the exact model version described in the paper or uploaded to the supplement, unless this is impossible for reasons beyond the control of authors. All papers must include a section, at the end of the paper, entitled "Code availability". Here, either instructions for obtaining the code, or the reasons why the code is not available should be clearly stated. It is preferred for the code to be uploaded as a supplement or to be made available at a data repository with an associated DOI (digital object identifier) for the exact model version described in the paper. Alternatively, for established models, there may be an existing means of accessing the code through a particular system. In this case, there must exist a means of permanently accessing the precise model version described in the paper. In some cases, authors may prefer to put models on their own website, or to act as a point of contact for obtaining the code. Given the impermanence of websites and email addresses, this is not encouraged, and authors should consider improving the availability with a more permanent arrangement. Making code available through personal websites or via email contact to the authors is not sufficient. After the paper is accepted the model archive should be updated to include a link to the GMD paper.

As GitLab is not a persistent archive, please provide a persistent release for the exact source code version used for the publication in this paper. As explained in https://www.geoscientific-model-development.net/about/manuscript_types.html the preferred reference to this release is through the use of a DOI which then can be cited in the paper. For projects in GitHub a DOI for a released code version can easily be created using Zenodo, see https://guides.github.com/activities/citable-code/ for details.

Yours, Astrid Kerkweg

---

## Author Comment (AC1) · 21 Dec 2020

Dear executive editor,

Please note that the MAR source code is actually already included in the persistent archive provided through zenodo (see the sentence "However, we have also archived a specific version of the dataset used in this study together with the MAR source code and outputs for replication of this study at https://zenodo.org/record/4314872" in the Code and data availability section). The GitLab link is provided in addition to the zenodo repository as it is the platform through which we currently release our model different versions. I will make sure in the revised version of the manuscript that the code

availability through GitLab is clearly exposed as an alternative to the zenodo repository.

Best wishes Charles Amory

---

## Referee Comment (RC1) · Anonymous Referee #1 · 2 Feb 2021

This paper presents recent developments of the regional climate model MAR to simulate drifting and blowing snow processes over ice sheets and a detailed evaluation of the model over Adelie Land, East Antarctica. The authors propose several modifications to the blowing snow module implemented in MAR that affect the computation of the threshold wind speed for the initiation of snow transport and the simulation of the evolution of surface snow properties during blowing snow events. A configuration of MAR at 10-km grid spacing is then used to simulate the drifting snow climate and surface mass balance in Adelie Land from 2004 to 2018. Simulations are evaluated in terms of near-surface atmospheric variables (wind speed and direction, air temperature and relative humidity), blowing snow variables (occurrence, mass flux) and

surface mass balance. This evaluation reveals a good ability of MAR to reproduce the occurrence of blowing snow events and their intensity at two locations. The authors also show improved performances compared to an older version of the model, despite differences in the model configuration between the two experiments. An interesting sensitivity analysis (Sec. 6.2) highlights the importance of accounting for the evolution of the properties of surface snow during blowing snow events and its feedback on blowing snow occurrence.

The paper is well written, and the results are well described. The subject of this study is interesting for a large community studying the surface mass balance of ice sheets. Prior to publication in GMD, certain aspects of the methods and results warrant further discussions and clarifications. They are listed below as general comments and are followed by more specific and technical comments. I recommend the manuscript to be accepted subject to the revisions outlined below.

General comments

1. The main purpose of this paper is to present a new version of the drifting-snow scheme implemented in MAR. This scheme is well described in the paper and is extensively evaluated. In this context, it would be very valuable for the reader to be able to clearly assess the benefit of this new drifting snow scheme compared the previous version. So far, the author present in Sect. 6.1 a comparison with previous simulations made with MAR but they differ in terms of horizontal resolution and lateral boundary conditions. Would it be possible to present a clean comparison between the two drifting snow schemes? There is certainly no need to run the whole period but 2 years could certainly be selected as done in Sect 6.2. Such comparison would certainly confirm the results shown on Fig 11 and highlight the benefit of using the new drifting-snow scheme implemented in MAR.

2. The authors explain at P6 L 152-153 that the drifting snow module in MAR uses a formulation for the particle ratio in the saltation layer taken from Pomeroy and Gray

(1990) and used in Bintanja (2000a). However, in Eq. 5 of the manuscript, the author uses for the saltation efficiency the formulation $e\_salt = 1/(3.25\ u\_star)$. This formulation is not consistent with Pomeroy and Gray (1990, page 1587) and Bintanja (2000). Indeed, in these two papers, the saltation efficiency is written as $e\_salt = 0.68/(c\_t * u\_star) = 1/(4.2 * u\_star)$ with $c\_t = 2.8$. The formulation given in Eq. 5 is consistent with Pomeroy (1989) as explained in the paper describing the initial drifting snow module in MAR (Gallee et al., 2001). The author should clarify this point and if they are using the formulation given by Eq. 5, I recommend them to use the formulation of Pomeroy and Gray (1990) which can be considered as the reference for the Pomeroy saltation module.

The formulation for the mixing ratio in the saltation layer detailed in Eq 5. gives a maximal value of the mixing ratio for intermediate values of the friction velocity. This is a potential limitation of this formulation that has been identified by Bintanja (2000a). Bintanja (2000b) tested another formulation where the mass concentration in the saltation layer increases monotonically with the friction velocity (see Eq. 2 in Bintanja (2000b)). Did the author test this approach in their model and how does it compare with the flux measurements collected at D17 and D47? I recommend the authors to add in their manuscript an additional sensitivity experiment and a related discussion about the validity of this formulation and its potential limitation.

Specific comments

Abstract L1: I understand that the authors want to use the term "drifting snow" throughout the paper as mentioned in the first paragraph of the introduction. However, I recommend being more general in the abstract and to use the term "drifting and blowing snow" to avoid any confusion for the reader who would start by reading the abstract.

P 4 L 95-96: the initialization of the snow-related variables can certainly be presented in Sect. 2.2 focusing on the snowpack model. The authors should also detail what they are using as initial conditions for the variables describing the snow microstructure.

P 7 L 183-184: the drifting snow module in MAR treats drifting snow particles in the atmosphere using the cloud microphysical scheme of MAR despite differences of size and shape between drifting snow particles and cloud snow particles (e.g. Nishimura and Nemoto, 2005). Could the author briefly discuss how this assumption affect the simulation of sublimation?

P 8 L 229-230: the wind-tunnel results of Sommer et al (2018) suggest that wind hardening is only found where snow has been deposited. Based on this result, I recommend the authors to reformulate their sentence to explain that their parametrisation aims at representing the post-deposition increase of the grid-average density in a grid cell exposed to snow transport.

P 8 Sect. 3.5: the authors should detail at which step is treated the deposition flux (sedimentation) from the atmosphere to the surface. My understating is that it happens at step 5, but I think it could be more specific.

P 9 L 249-250: Only the surface snow layer can be eroded in the new drifting snow implemented in MAR. It is not clear why the erosion of the underlying layer is not allowed if the surface layer is totally removed during one time step. Is this choice motivated by technical reasons to simplify the code or is there another motivation?

P 11 L 313: The FlowCapt sensors are sensitive to the impact of snow particles on the tube. Drifting snow particles in the saltation and suspension layers will therefore influence the value measured by the FlowCapts. However, the mass flux in the saltation layer is ignored in Eq. 10 when computing the value of the modelled drifting snow flux that is compared with the FlowCapt value. Can the author justify their choice? Could it be a reason for the underestimation of observed near-surface drifting snow transport shown on Fig. 6?

P 12 Fig. 1: how is defined surface snow density since the thickness of the surface snow layer is evolving with time?

P 13 L 361: Does MAR includes a parameterization for wind gusts, and could it be evaluated using the wind speed data available at D17 and/or D47?

L 19 Sect 5.4: In this section, the authors evaluate the mean annual surface mass balance (Fig. 9). Did they also consider evaluating the ability of MAR to reproduce the inter-annual variability of surface mass balance and how does it compare with the ability of the model to simulate the inter-annual variability of the drifting snow transport?

L 23 L 485: the sensitivity analysis presented in Sect 6.2 illustrates well how the evolution of the properties of surface snow during drifting snow events influences the frequency of these events and the amount of snow transported during these events. The authors test three values for the density of fresh snow ranging between 250 and 350 kg m-3. These values are much larger than the typical values for fresh snow used for snow model in mountainous environments (e.g. Helfricht et al., 2018). It is clear that the polar and mountainous environments differ but it would be interesting if the authors could better justify their choice for the range of values of fresh snow considered in this study. To what extent, are these values already partially integrating the effect of fragmentation on wind-blown snowflakes (Comola et al., 2017)? And as a consequence, what is the recommendation of the authors to properly separate between a representative value for fresh snow density and a useful parametrization to handle post-depositional increase of snow density during drifting snow events?

P 26 L 518: Note that the parameterization of Vionnet et al. (2012) for the threshold wind speed has been mainly used in applications of the Crocus snowpack model in polar environment. In alpine context (e.g., Vionnet et al., 2014), only the erodibility index given by Eq 3 of this manuscript is used.

Technical comments

Text

Abstract L2: it is not clear why the term "mostly" is used here. Drifting snow originates from particles raised from the surface of the snowpack (https://glossary.ametsoc.org/wiki/Drifting_snow).

P3 L 63-64: are there any references that support this affirmation about the performances of the former physical parameterization of drifting snow processes in MAR?

P5 L 125: I recommend the authors to mention here that the sensitivity to the formulation of the threshold wind speed for snow transport is quantified and discussed in Sect. 6.2.

P5 L 134: do the authors means something like "the wind speed at the lowest prognostic level of the model"? The formulation "... in the lowest model ..." is not clear.

P 12 L 343: replace m s-1 by m s$^{-1}$.

Figure

Figure 2: A density plot could be used to make the plots easier to read to the large numbers of points.

Figure 6: it is not easy to make the distinction between the points for D47 and D17. Maybe use two figures.

References

- Bintanja, R. (2000a). Snowdrift suspension and atmospheric turbulence. Part I: Theoretical background and model description. Boundary-layer meteorology, 95(3), 343-368.

- Bintanja, R. (2000b). Snowdrift suspension and atmospheric turbulence. Part II: Results of model simulations. Boundary-layer meteorology, 95(3), 369-395.

- Comola, F., Kok, J. F., Gaume, J., Paterna, E., & Lehning, M. (2017). Fragmentation of wind-blown snow crystals. Geophysical Research Letters, 44(9), 4195-4203.

- Gallée, H., Guyomarc'h, G., & Brun, E. (2001). Impact of snow drift on the Antarc-

tic ice sheet surface mass balance: possible sensitivity to snow-surface properties. Boundary-Layer Meteorology, 99(1), 1-19.

- Helfricht, K., Hartl, L., Koch, R., Marty, C., & Olefs, M. (2018). Obtaining sub-daily new snow density from automated measurements in high mountain regions. Hydrology and Earth System Sciences, 22(5), 2655-2668.

- Nishimura, K., & Nemoto, M. (2005). Blowing snow at Mizuho station, Antarctica. Philosophical Transactions of the Royal Society A: Mathematical, Physical and Engineering Sciences, 363(1832), 1647-1662.

- Pomeroy, J. W. (1989). A process-based model of snow drifting. Annals of Glaciology, 13, 237-240.

- Pomeroy, J. W., & Gray, D. M. (1990). Saltation of snow. Water resources research, 26(7), 1583-1594.

- Sommer, C. G., Lehning, M., & Fierz, C. (2018). Wind tunnel experiments: influence of erosion and deposition on wind-packing of new snow. Frontiers in Earth Science, 6, 4.

- Vionnet, V., Brun, E., Morin, S., Boone, A., Faroux, S., Moigne, P. L., ... & Willemet, J. M. (2012). The detailed snowpack scheme Crocus and its implementation in SURFEX v7. 2. Geoscientific Model Development, 5(3), 773-791.

- Vionnet, V., Martin, E., Masson, V., Guyomarc'h, G., Naaim-Bouvet, F., Prokop, A., ... & Lac, C. (2014). Simulation of wind-induced snow transport and sublimation in alpine terrain using a fully coupled snowpack/atmosphere model. The Cryosphere, 8(2), 395-415.

---

## Referee Comment (RC2) · Anonymous Referee #2 · 23 Feb 2021

It was a pleasure to read and review this manuscript. The model development work presented in this manuscript is very thorough and of high quality. I think that it is well suited for GMD, since it concerns the description of the implementation of drifting snow physics in a widely used atmospheric model (MAR). It is likely that the broader scientific community benefits from the improvements made to the model. The implemented drifting snow physics capture most known processes, including sublimation and describes the compaction effect of the surface firn layer during drifting snow. The new drifting snow physics provides a very good agreement with drifting snow fluxes in terms of timing and amount. Additionally, MAR captures SMB gradients in the coastal region well. I can recommend publication after minor revisions, listed below.

[Figure]

Some broader comments:

- I wonder how the statistics would look like when drifting snow events are separated in precipitation and non-precipitation periods (for example based on the 100 m particle concentration, as described in L224). That could show to what extent the description of the firn layer can really accurately predict erosion and drifting snow. I can imagine that cases with precipitation from the atmosphere poses less of a challenge for the model than eroding snow from the firn layer.

- Another drawback is that the surface density was not validated using field observations, which may not be available, but it's not so clear to what extent the chosen description is rather pragmatic, simply to serve the drifting snow physics or if it matches actual firn densities in the upper firn layers. If authors have such measurements available (like the ones presented in Figure S2), it could be a valuable addition (for example by adding MAR simulated surface density to Figure S2 in the supplement), but I don't consider it essential for publication of the manuscript.

- It's demonstrated that the drifting snow is simulated more accurately, but the good agreement for SMB is not compared to earlier model versions. It could be an addition to Fig. 9 to show results from previous versions of MAR.

Specific comments:

- Abstract L12/13: I would consider the statement that the MAR drifting-snow physics can serve as a basis for other models maybe a bit prematurely, since I think it would be important that the surface firn properties are validated (particularly density) against ground-truth.

- Introduction: some statements could be a bit expanded upon and made more concrete:

L20 - Maybe add a quantitative amount of sublimation found by the cited studies

L43/44 - "Arbitrary adjustments of model parameters favouring one can be made at the

expense of the other (e.g., van Wessem et al., 2018)" I suggest to briefly summarize what they found.

L55/56 - "from their numerous interactions with the atmosphere and the snow surface organized in a complex system of positive and negative feedback mechanisms" Please expand what the cited studies found in this regard.

- Introduction L40-42: I think it is also important to realize here that the need to explicitly describe drifting snow processes also increases with the tendency towards finer meshes of the atmospheric models used to study Antarctic SMB.

- Section 2.1: I always prefer that the reader get some information about computational efforts. A simple sentence can be sufficient, for example that running MAR over the 15 years on the 80x80 grid cells took XXX CPU hours, or something similar.

- Section 2.1., L97: How I interpret this sentence is that one simulation was run from 1994-2004, and that the firn state in 2004 then served as a basis for all the other simulations (including the sensitivity study). Is that true? Maybe make this explicit.

- MAR has been used before with the snow cover model CROCUS (Vionnet et al. (2012), https://gmd.copernicus.org/articles/5/773/2012/gmd-5-773-2012.pdf). Maybe section 2.2 should detail why instead of CROCUS, SISVAT was used. It's not so clear to me since apparently CROCUS is part of SISVAT, or some routines of CROCUS are used by SISVAT? Particularly, why is SISVAT more suitable than CROCUS for modeling drifting and blowing snow?

- Section 3 could benefit from a few introductory sentence of how it is structured. I was surprised for example that the section "Initiation of drifting snow" did not describe how snow was eroded from the firn layer. Maybe rename to "Threshold friction velocity for initiation of erosion". Currently, it's a bit difficult to understand the logic between the different subsections.

- Section 3.1: I think this section already needs to refer to Appendix B.

I know that Eq. 1-4 have been published before (Gallee et al., 2001), but I noticed that Eq. 3 corresponds to the "fresh snow" category in Eq. 1 in Gallee et al. (2001). However, it is commonly known that the snow surface in Antarctica can consist of old snow (see for example Picard et al. (2019)). What is the rationale that here, only the fresh snow category is used?

Section 3.2: L178/179 discusses the upward surface flux, but I understand that this is the flux from the saltation layer (which is not explicitly treated by the dynamical core of MAR) to the suspension layer. So when I understand correctly, there are three components: the firn layer, the saltation layer (both not considered by the dynamical core), and the suspension layer, which is from the lowest model layer in MAR upward. I assume Eq. 6 then describes the flux between the saltation layer and the suspension layer. I can recommend a sketch here to better illustrate this. A schematic sketch would probably improve the readability of Section 3.

Eq. 5: It's not clear how the units are treated here. All units are declared following meter and second, yet $q\_salt$ is expressed as kg/kg. Does this mean that some conversions using density is missing from the equation?

Section 3.3: Can the authors derive any quantification of the sublimation of drifting snow from their simulations? L191: Does the particle absorption of solar radiation increase sublimation? Can such information derived from the model simulations?

L206-210, and L249-250: What is the rationale for restricting erosion to the surface layer only? The original approach in MAR intuitively makes more sense, where the firn can erode until the mass flux is satisfied, or the snow is too dense/bonded to be erosion.

L254: At item 3: maybe explicitly discuss here the scenario that ER is positive (erosion) *and* the scenario that ER is negative (i.e., deposition).

L285: "further inland" is rather qualitative. Maybe add how many kilometers inland is

meant here.

L353: It's a little bit strangely formulated, since Fig. 3 only shows D17, not D47. So Fig. 3 is not really showing that the values are closer to observation than for D47.

L401-402: I don't comprehend how occurrences are missed at coarser temporal resolution. I assume that the coarser temporal resolution sums the mass fluxes over the coarser time steps, such that no information is lost?

L380-382: When the duration of events is underestimated, one would also expect an underestimation of total mass flux in events. It seems a bit in contradiction with what is argued later (L416/417) that the main events are correctly simulated and that the underestimation stems from particularly the low wind speed events. I actually think that there is also quite some uncertainty from the simulated firn properties, as mentioned in L428/429.

Fig. 9: It could be a nice addition to show the elevation or terrain slope angle along the transect here as well. It looks like that the terrain gets steeper near the coast and may also exhibit more variability. That variability probably drives SMB variability (as for example shown in Dattler et al. (2019)).

L588: "Both parameterizations are given for rho_s = 450 kg/mˆ3" I don't comprehend this sentence, since the functions B1-B3 are all using a variable rho_s?

Technical corrections:

- Two comments to make the abstract better comprehensible: L6: I suggest "drifting-snow compaction of the uppermost firn layer." L7/8: I suggest "and a rewrite of the parameterization for the threshold friction velocity, above which snow erosion initiates".

- L15/16: I suggest "wind-driven ablation or accumulation", since that's in better line with the discussion in the first paragraph.

- Fig. 1: the red labels on purple background are very difficult to see, and definitely

not easy for people with eye-sight problems / color blindness. Maybe put a white box behind the label, or improve the figure otherwise.

- L134: word missing after "lowest model".

- L155: even though pretty obvious, I recommend to add the value taken for gravitational acceleration.

- L239: I suggest explicitly referring to Eq. 1.

- L397: "an estimation"

- L475: "Improvements ... are illustrated"

- Fig. 7: is the horizontal axis the observed or simulated wind speed?

- I suggest to incorporate Appendix A in the main text.

---

## Author Comment (AC2) · 31 Mar 2021

We thank reviewer RC1 for making relevant and constructive comments that will help to improve our manuscript. Our responses are reported below in blue.

This paper presents recent developments of the regional climate model MAR to simulate drifting and blowing snow processes over ice sheets and a detailed evaluation of the model over Adelie Land, East Antarctica. The authors propose several modifications to the blowing snow module implemented in MAR that affect the computation of the threshold wind speed for the initiation of snow transport and the simulation of the evolution of surface snow properties during blowing snow events. A configuration of MAR at 10-km grid spacing is then used to simulate the drifting snow climate and surface mass balance in Adelie Land from 2004 to 2018. Simulations are evaluated in terms of near-surface atmospheric variables (wind speed and direction, air temperature and relative humidity), blowing snow variables (occurrence, mass flux) and surface mass balance. This evaluation reveals a good ability of MAR to reproduce the occurrence of blowing snow events and their intensity at two locations. The authors also show improved performances compared to an older version of the model, despite differences in the model configuration between the two experiments. An interesting sensitivity analysis (Sec. 6.2) highlights the importance of accounting for the evolution of the properties of surface snow during blowing snow events and its feedback on blowing snow occurrence.

The paper is well written, and the results are well described. The subject of this study is interesting for a large community studying the surface mass balance of ice sheets. Prior to publication in GMD, certain aspects of the methods and results warrant further discussions and clarifications. They are listed below as general comments and are followed by more specific and technical comments. I recommend the manuscript to be accepted subject to the revisions outlined below.

General comments
1. The main purpose of this paper is to present a new version of the drifting-snow scheme implemented in MAR. This scheme is well described in the paper and is extensively evaluated. In this context, it would be very valuable for the reader to be able to clearly assess the benefit of this new drifting snow scheme compared to the previous version. So far, the authors present in Sect. 6.1 a comparison with previous simulations made with MAR but they differ in terms of horizontal resolution and lateral boundary conditions. Would it be possible to present a clean comparison between the two drifting snow schemes? There is certainly no need to run the whole period but 2 years could certainly be selected as done in Sect 6.2. Such comparison would certainly confirm the results shown on Fig 11 and highlight the benefit of using the new drifting-snow scheme implemented in MAR.

The results presented in Sect. 6.1 and used for comparison have been produced with a former version of the model (v2, developed at the University of Grenoble) which has started diverging with the version used in this study (v3, developed at the University of Liège) more than 15 years ago. Version v2 is unfortunately not maintained anymore and the source code has even not been conserved.
As nothing had ever been published with MARv3 regarding drifting-snow applications the only option we had to highlight improvements within the drifting-snow scheme of the newest version was to rely on previously published work with MARv2 for January 2011, for which only the results for the reproduction of the corresponding paper (Amory et al., 2015) had been preserved. However, the very poor performance of MARv2 in simulating the drifting-snow mass flux as illustrated in Fig. 11 for a few events in January 2011 suggests that a comparison over a longer time period would have likely given a quite similar picture.
We agree though that a more comprehensive evaluation of MARv2 would have required working on similar periods with a similar model setup (number of vertical levels, horizontal resolution, lateral boundary conditions) and focusing on surface mass balance as well to help quantify benefits of using the new drifting-snow scheme implemented in MAR without any ambiguity. Since MARv2 has never been employed for surface mass balance applications, this comparison is however of no consequence for previously published works with that version of the model.

2. The authors explain at P6 L 152-153 that the drifting snow module in MAR uses a formulation for the particle ratio in the saltation layer taken from Pomeroy and Gray (1990) and used in Bintanja (2000a). However, in Eq. 5 of the manuscript, the author uses for the saltation efficiency the formulation

e_salt = 1/ (3.25 u_star). This formulation is not consistent with Pomeroy and Gray (1990, page 1587) and Bintanja (2000). Indeed, in these two papers, the saltation efficiency is written as e_salt = 0.68 / (c_t * u_star) = 1/ (4.2 * u_star) with c_t = 2.8. The formulation given in Eq. 5 is consistent with Pomeroy (1989) as explained in the paper describing the initial drifting snow module in MAR (Gallee et al., 2001). The author should clarify this point and if they are using the formulation given by Eq. 5, I recommend them to use the formulation of Pomeroy and Gray (1990) which can be considered as the reference for the Pomeroy saltation module.

The formulation for the mixing ratio in the saltation layer detailed in Eq 5. gives a maximal value of the mixing ratio for intermediate values of the friction velocity. This is a potential limitation of this formulation that has been identified by Bintanja (2000a). Bintanja (2000b) tested another formulation where the mass concentration in the saltation layer increases monotonically with the friction velocity (see Eq. 2 in Bintanja (2000b)). Did the author test this approach in their model and how does it compare with the flux measurements collected at D17 and D47? I recommend the authors to add in their manuscript an additional sensitivity experiment and a related discussion about the validity of this formulation and its potential limitation.

Thank you for pointing out this mistake. We have corrected the reference made in the text and the formulation of Pomeroy and Gray (1990) will be adopted in the next model version. We have verified that no significant differences in the near-surface drifting-snow mass flux occur by adjusting the formulation for qsalt (see more details below).

[Figure]

Figure R1. Evolution of qsalt as a function of u* for the parameterizations of Pomeroy (1989), Pomeroy and Gray (1990) and Bintanja (2000b), respectively indicated as P89, PG90 and B00.

Figure R1 illustrates that a significant sensitivity to the parameterization of the saltating mass flux could be expected with the alternative proposed by Bintanja (2000b), especially for u* > 0.6 m s [-1] from which the formulation starts to diverge from the two others. However, the comparison of the snow mass fluxes resulting from the two formulations of PG90 and B00 with the control run P89 shows very small differences (< 2% ) in the cumulative snow mass transport at D47 and D17 and similar drifting-snow

frequency values (see Table 4 in the revised version). This is caused by the limitation made on the erosion of only the mass available in the surface layer per time step (see our response to specific comment #6), which is strong enough to reduce the sensitivity to the formulation of qsalt.
We have summarized this finding in a paragraph that has been added to Sect. 6.2 and Fig. R1 is proposed in the supplement.

Specific comments
1. Abstract L1: I understand that the authors want to use the term "drifting snow" throughout the paper as mentioned in the first paragraph of the introduction. However, I recommend being more general in the abstract and to use the term "drifting and blowing snow" to avoid any confusion for the reader who would start by reading the abstract.
Many authors have referred to drifting and/or blowing snow for describing different processes (saltation, combined or not with suspension, wind-driven snow transport > 2 m and/or < 2 m, local erosion combined or not with horizontal advection, etc..) leading to a potential confusion of the actual meaning of this term when not properly defined.
We understand your concern about using more generic wording in the abstract. The paper has however been conceived around an explicit definition of drifting snow (as explained in the introduction). Including blowing snow in the first sentence of the abstract might possibly introduce confusion because of several instances of drifting snow declined in compound words such as drifting-snow scheme, drifting-snow physics, drifting-snow frequency, drifting-snow event etc (that would also include blowing snow in a different, more general meaning). For a matter of consistency of the abstract with the rest of the paper, rather we suggest to specify the height of particles transport we refer to: "*Drifting snow, or the wind-driven transport of cloud- and surface-originating snow particles below and above 2 m above ground [...]*".

2. P 4 L 95-96: the initialization of the snow-related variables can certainly be presented in Sect. 2.2 focusing on the snowpack model. The authors should also detail what they are using as initial conditions for the variables describing the snow microstructure.
The following paragraph has been added to Sect. 2.2:
"*The snowpack was uniformly initialised with snow grain shape parameters of fresh snow (see Sect. 3.3 for definition) and a density of 500 kg m^-3 assuming a null water liquid content. The initial surface snowpack temperature is set to the reanalysis near-surface air temperature and then discretised along a predefined layer-thickness profile as a function of distance to the surface to determine the temperature of internal snowpack layers. The model was then run from 1994 so that the snowpack had reached equilibrium with the climate preceding the period of interest (2004-2018) after a spin-up time of 10 years.*".

3. P 7 L 183-184: the drifting snow module in MAR treats drifting snow particles in the atmosphere using the cloud microphysical scheme of MAR despite differences of size and shape between drifting snow particles and cloud snow particles (e.g. Nishimura and Nemoto, 2005). Could the author briefly discuss how this assumption affect the simulation of sublimation?
All the following information is now detailed in the manuscript in Sect. 3.4.:
Phase changes of atmospheric water species in MAR are resolved within each atmospheric level according to the microphysical processes described in Lin et al. (1983). In particular, sublimation (their Eq. (31), p. 1072) is calculated by assuming an exponential size distribution of suspended (cloud and eroded) snow particles (Gallée, 1995)

$$n_s = n_0 \exp(-\lambda_s * D_s)$$

with $n_s$ the number of snow particles of diameter $D_s$ per unit volume, $n_0$ an empirical constant that corresponds to the intercept parameter of the size distribution, and $\lambda_s$ the dispersion parameter

$$\lambda_s = (\pi * \rho * n_0 / \rho_a * q_s)^{\wedge}(\tfrac{1}{4})$$

where rho is the snow particle density set to 100 kg m⁻³, rho_a is the air density and qs is the snow particle ratio (kg/kg). Snow particles are considered as graupel-like snow of hexagonal type and the spectrally-averaged snow particle diameter Ds is prescribed as a constant following Locatelli and Hobbs (1974).

Not distinguishing on the origin of particles despite differences in shape and size between cloud and eroded snow particles (Nishimura and Nemoto, 2005) can affect the estimation of sublimation according to the predominance of one type of particles over the other in the actual airborne snow mass. An overestimation of atmospheric sublimation rates within drifting-snow layers mainly consisting of eroded particles can be expected, which would be partially counterbalanced by the increased negative feedback of sublimation and all the less pronounced as the relative contribution of cloud particles prevails.
Note that, despite the prescription of a constant snow particle diameter, the seasonal cycle in air relative humidity at 2 m is well captured by the model (see Le Toumelin et al. (2020) - their Fig. 4a).

4. P 8 L 229-230: the wind-tunnel results of Sommer et al (2018) suggest that wind hardening is only found where snow has been deposited. Based on this result, I recommend the authors to reformulate their sentence to explain that their parametrization aims at representing the post-deposition increase of the grid-average density in a grid cell exposed to snow transport.
We have reformulated the first sentence as "*The post-deposition increase in snow density through wind hardening is accounted for in the model by increasing the grid-average density of the uppermost snowpack layer in each grid cell exposed to drifting snow*".

5. P 8 Sect. 3.5: the authors should detail at which step is treated the deposition flux (sedimentation) from the atmosphere to the surface. My understanding is that it happens at step 5, but I think it could be more specific.
We have reformulated the description of step 5 to improve clarity: "*The drift fraction is obtained from Eq. (8). Snow is deposited at the surface and surface density is adjusted according to Eq. (7)*".

6. P 9 L 249-250: Only the surface snow layer can be eroded in the new drifting snow implemented in MAR. It is not clear why the erosion of the underlying layer is not allowed if the surface layer is totally removed during one time step. Is this choice motivated by technical reasons to simplify the code or is there another motivation?
The main motivation is indeed to i) avoid numerical instability related to the erosion of multiple layers that hinder the surface temperature from reaching equilibrium and ii) prevent significant rearrangements of the snowpack per time step increasing the computational cost. Furthermore this choice is also related to the discretisation of the snowpack, which is not directly based on mass but on layer thickness. The maximum thickness of the surface layer is set to 2 cm and the model timestep is 60 s. With a maximum surface density (for an erodible layer) of 450 kg m⁻³, removing the entire layer within one time step leads to an erosion rate of ~1 cm min⁻¹, which can be thus regarded as an upper bound. While no observations are currently available to assess this limit, we consider this hypothesis as acceptable since it enables a good agreement between observed and simulated snow mass fluxes.
The motivation for this choice is now better explained in the corresponding paragraph: "*A different feature of the current drifting-snow scheme of MAR contrasting with earlier versions is that, instead of being simultaneously distributed over several upper snow layers, the influence of erosion and deposition at each model time step (60 s) is restricted to the uppermost snow layer only, under the consideration that only the surface snowpack layer can exchange momentum and mass with the atmosphere. For deposition, this reduces the computational cost by preventing rearrangements of several snow layers per time step. For erosion, this avoids numerical instabilities related to the likely removal of several snow layers deeper in the snowpack while the computation of the surface temperature and energy balance is based on the surface layer only. Snow layers with different characteristics may thus be deposited or exposed successively at the top of the snowpack during a drifting-snow event, thus influencing the simulated surface albedo.*".

7. P 11 L 313: The FlowCapt sensors are sensitive to the impact of snow particles on the tube. Drifting snow particles in the saltation and suspension layers will therefore influence the value measured by the

FlowCapts. However, the mass flux in the saltation layer is ignored in Eq. 10 when computing the value of the modelled drifting snow flux that is compared with the FlowCapt value. Can the author justify their choice? Could it be a reason for the underestimation of observed near-surface drifting snow transport shown on Fig. 6?

The mass removed from the surface through erosion is expressed by the turbulent flux of surface snow particles (Eq. 6 of the manuscript) which indeed corresponds, from a theoretical perspective, to the snow mass suspended from the saltation layer. However, since we want to account for the actual airborne snow mass (including erosion) to calculate the snow mass flux that is actually simulated by the model, saltation needs to be excluded from the calculation as it is not explicitly resolved by MAR and only serves as a lower boundary condition for the suspension layer. This numerical approach could be responsible for a misrepresentation of the observed near-surface drifting-snow transport as shown on Fig. 6, but more generally the ability of the model in representing drifting snow results from the balance between all the factors influencing the airborne snow mass, including other choices (and possibly even more influential) made in the microphysical, turbulence and surface schemes.

8. P 12 Fig. 1: how is defined surface snow density since the thickness of the surface snow layer is evolving with time?

Surface snow density corresponds to the density of the uppermost layer of the snowpack. This is now specified in the figure caption.

9. P 13 L 361: Does MAR include a parameterization for wind gusts, and could it be evaluated using the wind speed data available at D17 and/or D47?

The source code of MAR indeed includes initial developments for a parameterization of wind gusts. However as it is an off-line procedure that still requires development, we think that it may lie beyond the scope of our study, especially given that observation data are only available at the half-hourly timescale.

10. L 19 Sect 5.4: In this section, the authors evaluate the mean annual surface mass balance (Fig. 9). Did they also consider evaluating the ability of MAR to reproduce the inter-annual variability of surface mass balance and how does it compare with the ability of the model to simulate the inter-annual variability of the drifting snow transport?

Figure R2 shows the comparison between observed and modelled annual SMB for the period 2004-2018 following the same methodology described in the paper. The analysis year by year illustrates the ability of MAR to reproduce the inter-annual variability of the SMB conjointly with drifting-snow transport. (Figs 5,6 in the initial manuscript). Similar conclusions than for the analysis of the mean annual SMB can be drawn here: MAR underestimates the SMB over the first tens of kilometers of the transect by locally simulating a persistent ablation zone, and better captures the SMB gradient further inland.

[Figure]

Figure R2. Simulated (squares, red solid curve) vs. observed (circles, blue dotted curve) annual SMB from 2004 to 2018 along the stake transect. Observed annual SMB values are averaged on MAR grid cells with no interpolation nor weighting. The spatial resolution is 10 km. Distance along the transect starts at the coast, and is computed as the average distance of all annual observations contained in each grid cell. The vertical dashed bars represent one spatial standard deviation of the observations.

11. L 23 L 485: the sensitivity analysis presented in Sect 6.2 illustrates well how the evolution of the properties of surface snow during drifting snow events influences the frequency of these events and the amount of snow transported during these events. The authors test three values for the density of fresh snow ranging between 250 and 350 kg m-3. These values are much larger than the typical values for fresh snow used for snow model in mountainous environments (e.g. Helfricht et al., 2018). It is clear that the polar and mountainous environments differ but it would be interesting if the authors could better justify their choice for the range of values of fresh snow considered in this study. To what extent, are these values already partially integrating the effect of fragmentation on wind-blown snowflakes (Comola et al., 2017)? And as a consequence, what is the recommendation of the authors to properly separate between a representative value for fresh snow density and a useful parametrization to handle post-depositional increase of snow density during drifting snow events?

In addition to the different environmental conditions between polar and mountainous regions that could justify prescription of different values for fresh snow density, the value of 300 kg m^-3 for the simulated fresh snow density in polar regions is frequently assumed (e.g., Lenaerts et al., 2012, Fausto et al., 2018) in the absence of a sophisticated firn compaction model. In this context, MAR is no exception and the value chosen for the density of fresh snow $rho\_0$ in the model is a compromise between the simplified representation of the snow microstructure and firn compaction, and the role also played by $rho\_0$ in the parameterization of the threshold friction velocity $u*t$ (Eq. 1 and drifting-snow compaction rate Eq. 9). We could indeed prescribe a fresh snow density value at deposition that differs from the reference density value used in Eq. (1) and that would be more in line with observations or theoretical considerations, but we wanted to preserve the consistency between the two parameterizations (Eqs. 1 and 9) and keep the current definition of $rho\_0$. This notably requires adopting values for $rho\_0$ that are probably above typical values for fresh snow as $u*t$ becomes a more restrictive criterion for erosion with decreasing $rho\_0$ (Fig. 12), and consequently that already partially integrate the effect of post-deposition processes. Ideally (and here is our recommendation) different approaches of higher complexity could be envisaged, for instance, by enlarging the set of snow microstructural properties accounted for (e.g., Lehning and Fierz, 2008) and removing the dependency to $rho\_0$ in the computation of $u*t$, prescribing fresh snow density as a function of atmospheric conditions (e.g., Vionnet et al., 2012), and/or including a more detailed representation of post-depositional processes including wind fragmentation, wind hardening during drifting snow, internal compaction and metamorphism. But all of this would require adaptation of the snowpack model in SISVAT and additional development experiences to properly account for and/or improve the representation of these processes. Our approach is much simpler and based on the current version of SISVAT accounting for its current level of complexity. In that sense, our choice made for the fresh snow density value is relative to the choices made in the drifting-snow model (in particular, to the current representation of $rho\_s$, $u*t$ as a function of $rho\_0$ and $rho\_s$, and drifting-snow compaction) but is undoubtedly perfectible. The sensitivity experiments on $u*t$ illustrate this idea of contextual dependency; changing $u*t$ for two other parameterizations found in the literature degrade the model performance. This does not mean that the two other parameterizations of $u*t$ are less recommendable but that the control parameterization of $u*t$ is more adapted to the current model version (including all the choices involved in parameterizing drifting snow). And most likely, so it is for the other parameterisations in their original development context.

The following sentence has been added to the text (P25 L534-536 in the revised version) when describing the sensitivity experiments on $u*t$ : "[…] *that result from the combined sensitivities to each of the three parameterisations. This notably requires adopting values for $rho\_0$ that are probably above typical values for fresh snow as $u*t$ becomes a more restrictive criterion for erosion with decreasing $rho\_0$ (Fig. 12), and consequently that already partially integrate the effect of post-deposition processes. Contrasting results between D47 and D17* […]*".*

12. P 26 L 518: Note that the parameterization of Vionnet et al. (2012) for the threshold wind speed has been mainly used in applications of the Crocus snowpack model in polar environments. In alpine context (e.g., Vionnet et al., 2014), only the erodibility index given by Eq 3 of this manuscript is used.

Thank you for this clarification. Note that the adaptation of the erodibility index for polar applications is also specified in Appendix B (or Appendix A of the revised version).

Technical comments

Text

Abstract L2: it is not clear why the term "mostly" is used here. Drifting snow originates from particles raised from the surface of the snowpack
(https://glossary.ametsoc.org/wiki/Drifting_snow).
The term « mostly » is used here to highlight the different types of particles that can contribute to the horizontal snow mass transport. Even if the definition found in the glossary of the AMS provides a theoretical context for describing drifting snow, in a natural environment drifting snow conditions are likely to involve particles raised from the surface mixed with particles directly originating from clouds, and in practice the application of this definition outside of clear-sky conditions (i.e., drifting particles only originating from the surface) would hardly account for the process as a whole. As an illustration, drifting snow has been shown to be frequently associated with low-pressure systems and concurrent precipitation in the Alps (Vionnet et al., 2013) and in coastal East Antarctica (Gossart et al., 2017) due to the increased availability of loose snow.
Moreover, cloud and eroded particles are accounted for in the snow mass flux estimates provided by both the FowCapt sensors and the model. While no clear consensus can be found around the semantics of drifting snow in the literature, in this study we thus chose to include the transport of cloud-originating particles in our definition of drifting-snow transport to guarantee the agreement between what, in our case, is measured by the instruments and simulated by the model. We tried to leave no ambiguity on that matter in the manuscript by referring to erosion when only the contribution of the surface to the snow mass flux is discussed and to drifting snow otherwise, as well as by referring to near-surface drifting-snow transport when quantities below a height of 2 m are described.
Nevertheless, we have adapted the first sentence of the abstract (see our response to specific comment #1) and completed the definition of drifting snow in the introduction: *"[...] erosion, deposition, horizontal and vertical transport of wind-driven cloud (i.e., that have not yet reach the surface) and eroded (raised from the surface) snow particles and their concurrent sublimation are all referred to as drifting-snow processes."*.

P3 L 63-64: are there any references that support this affirmation about the performances of the former physical parameterization of drifting snow processes in MAR?
This assertion only refers to unpublished preliminary experiments that served as a motivation for this study. We rephrased as "Unpublished *preliminary experiments with former physical parameterisations of drifting snow in MAR [...]"*.

P5 L 125: I recommend the authors to mention here that the sensitivity to the formulation of the threshold wind speed for snow transport is quantified and discussed in Sect. 6.2.
This has been added to the text.

P5 L 134: do the authors mean something like "the wind speed at the lowest prognostic level of the model"? The formulation ": : : in the lowest model : : :" is not clear.
This was a typo, thank you. We have reformulated as suggested.

P 12 L 343: replace m s-1 by m s$^{-1}$.
Corrected.

Figure

Figure 2: A density plot could be used to make the plots easier to read to the large numbers of points.
Thank you for this ingenious alternative. Figure 2 has been replaced with density scatter plots:

[Figure]

Figure R3. Density scatter plots of observed vs. simulated half-hourly wind speed (top), air temperature (middle) and air relative humidity with respect to ice (bottom) at 2 m height for stations D47 (left) and D17 (right). The coloured lines show the 1:1 line (dashed black) and the best linear fit (red).

Figure 6: it is not easy to make the distinction between the points for D47 and D17. Maybe use two figures.
Figure 6 now shows one panel for each station:

[Figure]

Figure R4. Scatter plots of observed vs. simulated near-surface drifting-snow transport for each drifting-snow event at D47 (red circles) and D17 (blue squares). As in Amory (2020a), a drifting-snow event is defined as a period over which the observed snow mass flux is above the detection threshold of 10^-3 kg m^-2 s^-1 for a minimum duration of 4 h.

References

Bintanja, R.: Snowdrift suspension and atmospheric turbulence. Part I: Theoretical background and model description, Boundary-LayerMeteorology, 95, 343–368, https://doi.org/10.1023/A:1002676804487, 2000a.

Bintanja, R.: Snowdrift suspension and atmospheric turbulence. Part II: Results of model simulations, Boundary-Layer Meteorology, 95,369–395, https://doi.org/10.1023/A:1002643921326, 2000b.

Comola, F., Kok, J. F., Gaume, J., Paterna, E., & Lehning, M. (2017). Fragmentation of wind-blown snow crystals. Geophysical Research Letters, 44(9), 4195–4203. https://doi.org/10.1002/2017GL073039.

Gallée, H.: 1995, Simulation of the Mesocyclonic Activity in the Ross Sea, Antarctica. Mon. Wea. Rev. 123, 2051–2069, https://doi.org/10.1175/1520-0493(1995)123<2051:SOTMAI>2.0.CO;2

Gossart, A., Souverijns, N., Gorodetskaya, I. V., Lhermitte, S., Lenaerts, J. T. M., Schween, J. H., Mangold, A., Laffineur, Q., and vanLipzig, N. P. M.: Blowing snow detection from ground-based ceilometers: application to East Antarctica, The Cryosphere, 11, 2,755–2,772, https://doi.org/10.5194/tc-11-2755-2017, 2017.

Helfricht, K., Hartl, L., Koch, R., Marty, C., and Olefs, M.: Obtaining sub-daily new snow density from automated measurements in high mountain regions, Hydrol. Earth Syst. Sci., 22, 2655–2668, https://doi.org/10.5194/hess-22-2655-2018, 2018.

Le Toumelin, L., Amory, C., Favier, V., Kittel, C., Hofer, S., Fettweis, X., Gallée, H., and Kayetha, V.: Sensitivity of the surface energy budget to drifting snow as simulated by MAR in coastal Adelie Land, Antarctica, The Cryosphere Discuss. [preprint], https://doi.org/10.5194/tc-2020-329, in review, 2020.

Lin Y.-L., Farley, R. D., and Orville, H. D.: 1983: Bulk Parameterization of the Snow Field in a Cloud Model. J. Appl. Meteorol. 22, 1065–1091, https://doi.org/10.1175/1520-0450(1983)022<1065:BPOTSF>2.0.CO;2

Locatelli, J. D., and P. V. Hobbs, 1974: Fall speeds and masses of solid precipitation particles. *J. Geophys. Res.*, 79, 2185–2197, https://doi.org/10.1029/JC079i015p02185.

Nishimura, K. and Nemoto, M.: Blowing snow at Mizuho station, Antarctica, Philos. T. R. Soc. A, 363, 1647–1662, https://doi.org/10.1098/rsta.2005.1599, 2005.

Pomeroy, J. W. and Gray, D. M.: Saltation of Snow, Water Resources Research, 26, 1,583–1,594, https://doi.org/10.1029/WR026i007p01583,1990.

Sommer, C. G., Wever, N., Fierz, C., and Lehning, M.: Investigation of a wind-packing event in Queen Maud Land, Antarctica, The Cryosphere, 12, 2,923–2,939, https://doi.org/10.5194/tc-12-2923-2018, 2018.

Vionnet, V., Guyomarc'h, G., Naaim Bouvet, F., Martin, E., Durand, Y., Bellot, H., Bel, C., and Puglièse, P.: Occurrence of blow-ing snow events at an alpine site over a 10-year period: Observations and modelling, Advances in Water Resources, 55, 53–63, https://doi.org/10.1016/j.advwatres.2012.05.004, 2013.

Vionnet, V., Martin, E., Masson, V., Guyomarc'h, G., Naaim-Bouvet, F., Prokop, A., Durand, Y., and Lac, C.: Simulation of wind-induced snow transport and sublimation in alpine terrain using a fully coupled snowpack/atmosphere model, The Cryosphere, 8, 395–415, https://doi.org/10.5194/tc-8-395-2014, 2014.

---

## Author Comment (AC3) · 31 Mar 2021

We thank reviewer RC2 for making a thorough analysis and interesting suggestions from which the manuscript will undoubtedly benefit. Our responses are reported below in blue.

It was a pleasure to read and review this manuscript. The model development work presented in this manuscript is very thorough and of high quality. I think that it is well suited for GMD, since it concerns the description of the implementation of drifting snow physics in a widely used atmospheric model (MAR). It is likely that the broader scientific community benefits from the improvements made to the model. The implemented drifting snow physics capture most known processes, including sublimation and describes the compaction effect of the surface firn layer during drifting snow. The new drifting snow physics provides a very good agreement with drifting snow fluxes in terms of timing and amount. Additionally, MAR captures SMB gradients in the coastal region well. I can recommend publication after minor revisions, listed below.

Some broader comments:

- I wonder how the statistics would look like when drifting snow events are separated in precipitation and non-precipitation periods (for example based on the 100 m particle concentration, as described in L224). That could show to what extent the description of the firn layer can really accurately predict erosion and drifting snow. I can imagine that cases with precipitation from the atmosphere poses less of a challenge for the model than eroding snow from the firn layer.

The fact that MAR generates drifting snow more easily with concomitant precipitation is inherent to the model itself: since the snow particle ratio qs contains the contribution of cloud particles and qs is used to compute the drifting-snow mass flux and determine drifting-snow occurrences, necessarily a combination of precipitation and wind results in drifting snow in the model. From this perspective, even by keeping the drifting-snow scheme switched off, MAR simulates (weak) transport of snow by the wind just by horizontal advection of snowfall during their residence into the atmosphere.

Indeed we could distinguish between precipitation and non-precipitation periods during drifting snow using for instance the ratio between the surface and 100 m particle concentrations, but the results would be sensitive to the threshold value used to determine a mixed drifting-snow event. By anticipating that, for a given value of the threshold ratio, we can show that drifting snow is better reproduced for mixed events, we could not assess if these events actually involve precipitation. In that case, we would need actual observations of precipitation to calibrate the threshold value and produce more robust results. Though this seems to be feasible for another locations in Antarctica where precipitation profiles are indeed available (Souverijns et al., 2018; Genthon et al., 2018), such observations are not available at D17 and D47 and we would rather keep this idea for another study.

- Another drawback is that the surface density was not validated using field observations, which may not be available, but it's not so clear to what extent the chosen description is rather pragmatic, simply to serve the drifting snow physics or if it matches actual firn densities in the upper firn layers. If authors have such measurements available (like the ones presented in Figure S2), it could be a valuable addition (for example by adding MAR simulated surface density to Figure S2 in the supplement), but I don't consider it essential for publication of the manuscript.

We agree that an evaluation of surface snow properties could be a lacking aspect in our evaluation exercise. Measurements of firn density in the upper firn layers are however quite limited in Antarctica, and this is all the more true for measurements in thin surface layers in Adelie Land, as would be required here. When available, density measurements are given for firn samples to 0.5 to 1 m in thickness that would not enable an evaluation of the simulated density of the surface layer bounded to a maximum of 0.02 m in thickness. To our knowledge, only the very few measurements presented in Fig. S2 would fit these requirements. These observations show surface density values around 200 kg m^-3 at the beginning of the drifting-snow episode, which are inevitably not captured by the model in which the minimum snow density value at deposition is set to 300 kg/m3 for practical purposes (see our response to comment #11 by RC1)..

In that sense, the chosen description of drifting-snow compaction does not necessarily enable a correspondence with actual snow surface densities, but rather merely serves the drifting-snow physics

to ensure a realistic time evolution of surface snow density and capture the associated feedback for snow erosion. The discussion on the pragmatic nature of this parameterisation has been included in the text: "*By fixing rho_0 and parameterising u\*t as an increasing function of rho_s (Eq. 1), Eq. (11) does not necessarily enable a correspondence with actual snow surface densities, but rather merely ensures a realistic time evolution of surface snow density. It also prevents large (positive) values of the difference u\*- u\*t to endure through time and thus acts as a negative feedback for snow erosion.*".

- It's demonstrated that the drifting snow is simulated more accurately, but the good agreement for SMB is not compared to earlier model versions. It could be an addition to Fig. 9 to show results from previous versions of MAR.
Excepted the results at D17 shown in Section 6.1, previous results with MARv2 are unfortunately not available anymore (see our response to general comment #1 of reviewer RC1) and have moreover never involved SMB products.

Specific comments:

- Abstract L12/13: I would consider the statement that the MAR drifting-snow physics can serve as a basis for other models maybe a bit prematurely, since I think it would be important that the surface firn properties are validated (particularly density) against ground-truth.
We have removed this sentence from the text and the abstract.

- Introduction:
some statements could be a bit expanded upon and made more concrete:
L20 - Maybe add a quantitative amount of sublimation found by the cited studies
This part of the introduction rather discusses the definition of atmospheric sublimation as an independent SMB term or not, without quantifying it. Moreover, the cited studies do not quantify it, they only describe it in terms of SMB from a different angle than the one considered here. We are currently working on the quantification of atmospheric sublimation in a fully dedicated forthcoming paper, which will surely be a much more appropriate context for such specifications.

L43/44 - "Arbitrary adjustments of model parameters favouring one can be made at the expense of the other (e.g., van Wessem et al., 2018)" I suggest to briefly summarize what they found.
van Wessem et al. (2018) evaluate the performance of RACMO2.3p2 (new version) compared to a former model version (RACMO2.3p1) in representing SMB and drifting-snow observations. The authors show that an improved representation of the SMB is obtained with the new model version by notably halving some saltation coefficient, efficiently halving the modelled snow mass transport vertically integrated over the whole drifting-snow layer, and reducing the agreement (from a positive to a negative bias) with observations when compared to the observed mass transport integrated over the first 2 meters above ground (see their Fig. 10).
However, we wish to keep this paragraph concise with an equivalent level of details for each reference in order to not lose the main focus of the paragraph, which is to comment on the linkages between SMB and drifting snow in a general modelling context. So we would rather keep the paragraph in its current version.

L55/56 - "from their numerous interactions with the atmosphere and the snow surface organized in a complex system of positive and negative feedback mechanisms" Please expand what the cited studies found in this regard
These studies were initially cited here as they both contain a description of part of the feedback mechanisms mentioned here (i.e., negative snow density feedback, surface roughness feedback, positive/negative buoyancy feedback) but do not discuss the model sensitivity to these feedbacks. A significant number of them (including those described in both publications) are accounted for in MAR and described along Sect. 3.3 in the initial version of the manuscript when appropriate. We hope to have clarified the sentence by rewriting: "*Numerical challenges associated with modelling drifting snow at the regional scale also arise from the numerous interactions of drifting-snow particles with the atmosphere and the snow surface organised in a complex system of positive and negative feedback*

*mechanisms. The difficulty involved in capturing the resulting strong non-linearity of drifting-snow processes depends on the representation and number of feedbacks accounted for (Gallée et al., 2013) and is mirrored through a high sensitivity of model results to parameter choices and significant discrepancies between simulated and observed snow mass fluxes (Lenaerts et al., 2014; Amory et al., 2015; van Wessem et al., 2018).".*

- Introduction L40-42: I think it is also important to realize here that the need to explicitly describe drifting snow processes also increases with the tendency towards finer meshes of the atmospheric models used to study Antarctic SMB.
Thanks for this relevant remark. We have added your comment at the end of the corresponding paragraph.

- Section 2.1: I always prefer that the reader get some information about computational efforts. A simple sentence can be sufficient, for example that running MAR over the 15years on the 80x80 grid cells took XXX CPU hours, or something similar.
We suggest the following complementary information: « *The time step is set to 60 s, for a computational cost of 72 CPU hours per year of simulation in the chosen configuration.* ».

- Section 2.1., L97: How I interpret this sentence is that one simulation was run from 1994-2004, and that the firn state in 2004 then served as a basis for all the other simulations (including the sensitivity study). Is that true? Maybe make this explicit.
The wording is indeed a bit clumsy. We have simply started the simulation 10 years before our period of interest (i.e., in 1994, or a spin-up time of 10 years) for the snowpack to reach a stable state.
We have rewritten the sentence to make it clearer: "*The model was then run from 1994 so that the snowpack had reached equilibrium with the climate preceding the period of interest (2004-2018) after a spin-up time of 10 years*".
For the sensitivity experiments, we only re-run the model from the simulation obtained with the control setup at the beginning of each year of investigation. This is now also explicitly mentioned for clarity: "*Simulated drifting-snow frequency and transport is evaluated for each experiment at site D47 for year 2010 and D17 for year 2013, restarting from the simulation obtained with the control setup.*".

- MAR has been used before with the snow cover model CROCUS (Vionnet et al. (2012), https://gmd.copernicus.org/articles/5/773/2012/gmd-5-773-2012.pdf). Maybe section 2.2 should detail why instead of CROCUS, SISVAT was used. It's not so clear to me since apparently CROCUS is part of SISVAT, or some routines of CROCUS are used by SISVAT? Particularly, why is SISVAT more suitable than CROCUS for modeling drifting and blowing snow?
The representation of snow in SISVAT was inspired from the developments made in CROCUS in its early version (early 90's) and significantly diverged later. The model presented in Vionnet et al. (2012) contains already much more sophisticated versions of the original routines from which SISVAT has been inspired. Similarly SISVAT is the original and current surface scheme of MAR and has also evolved with it. Today CROCUS and SISVAT are two different models that have been adapted to the needs of their users, so currently SISVAT is empirically (and naturally) more suitable than CROCUS for modelling snow transport with MAR. Significant differences exist between the two snow models and relate to different application contexts involving also compromises made on the computational cost (1-D, high-resolution simulation with CROCUS or local study case with the version coupled to the atmospheric model Meso-NH against coarser resolution, continent-wide investigations with MAR over climatological periods). Implementing the actual version of CROCUS in MAR would surely be an interesting work, that would however requires significant resources,developments, adaptations (for instance dendricity/sphericity in SISVAT vs specific surface area in CROCUS for the description of snow, refreezing accounted for in SISVAT and neglected in CROCUS), tests and reflexion on the level of sophistication required to optimize the simulations and preserve plausibility together with a reasonable computation time for 50 to 100 years of continent-wide simulation at tens of kilometres resolution.

- Section 3 could benefit from a few introductory sentence of how it is structured. I was surprised for example that the section "Initiation of drifting snow" did not describe how snow was eroded from the firn layer. Maybe rename to "Threshold friction velocity for initiation of erosion". Currently, it's a bit difficult to understand the logic between the different subsections.

Section 3.1 has been renamed as suggested, a schematic sketch is now provided in the manuscript (see Fig. R1) and an introductory paragraph as been added to the text:

*"This section describes the drifting-snow physics currently implemented in MAR. Details on the computation of the threshold friction velocity for snow erosion, snow-transport modes, interactions of drifting snow with the atmosphere and the surface, and then snow erosion and surface roughness are successively provided in the following subsections. A schematic sketch (Fig. 1) provides a general overview of the drifting-snow scheme."*.

[Figure]

*Figure R1. Schematic illustration of the drifting-snow scheme in MAR. Model variables are marked in bold black. The blue arrows denote mass and energy exchanges and drifting-snow processes are indicated in blue. The different computation steps listed in Sect. 3.5 are reported in red.*

- Section 3.1: I think this section already needs to refer to Appendix B.

Instead of referring to Appendix B (which gives details about the tested parameterisations for u*t not mentioned yet at this stage of the manuscript), and also following a suggestion made by reviewer RC1, we suggest to refer at the end of the Sect. 3.1 to the sensitivity analysis provided in Sect. 6.2 where a reference to Appendix B (or Appendix A in the revised version) is made.

- I know that Eq. 1-4 have been published before (Gallee et al., 2001), but I noticed that Eq. 3 corresponds to the "fresh snow" category in Eq. 1 in Gallee et al. (2001). However, it is commonly known that the snow surface in Antarctica can consist of old snow (see for example Picard et al. (2019)). What is the rationale that here, only the fresh snow category is used?

One motivation to simplify the parameterisation of u*t was to remove the discontinuity between the two members of the former parameterisation (or ensure continuity) over the range of modelled surface snow density values by generalizing the parameterisation for the fresh snow category to all snow categories. The discontinuity in the original version of Gallée et al. (2001) was indeed identified as a cause of instability during the development phase of the new version presented here, and the density value at which the switch from one member to the other was allowed appeared as a highly sensitive tuning parameter. Another objective, as explained in the text, was to minimise the dependency of u*t on variables for which virtually no information was available so that u*t depends only on surface density (similarly to what was done in Lenaerts et al. (2012) with RACMO and Liston et al., (2007) with SnowTran-3D). This is achieved in the model by prescribing a snow density at deposition which corresponds to a pseudo fresh-snow density and already partially accounts for the influence of postdepositional processes, taking into account the involvement of the fresh-snow density value rho_0 in the determination of u*t (see our response to comment #11 by RC1).

Although the current parameterisation of u*t corresponds to what has been conceptually described in Gallée et al. (2001) as the "fresh-snow" category, the contribution of old snow to u*t is accounted for by adjusting the surface snow density, which determines u*t, at each time step according to the proportion of drifting snow relative to fresh snow (Eq. 9 in the revised version). Surface layers mainly constituted of fresh snow are thus characterised by low density values, and thus lower u*t than less erodible layers of higher density including a higher proportion of older snow.

- Section 3.2: L178/179 discusses the upward surface flux, but I understand that this is the flux from the saltation layer (which is not explicitly treated by the dynamical core of MAR) to the suspension layer. So when I understand correctly, there are three components: the firn layer, the saltation layer (both not considered by the dynamical core), and the suspension layer, which is from the lowest model layer in MAR upward. I assume Eq. 6 then describes the flux between the saltation layer and the suspension layer. I can recommend a sketch here to better illustrate this. A schematic sketch would probably improve the readability of Section 3.

Your assumption is right. Saltation is not explicitly resolved by the model and the mass actually removed from the surface then corresponds to the upward mass exchange between the saltation and the suspension layers. This is now clearly mentioned in the text (Sect. 3.2) and a schematic sketch is provided in Fig. 1 in the revised version to better illustrate it (see Fig. R1).

- Eq. 5: It's not clear how the units are treated here. All units are declared following meter and second, yet q_salt is expressed as kg/kg. Does this mean that some conversions using density is missing from the equation?

The original formulation for qsalt (kg m^-3) is given in Pomeroy (1989) as

$$qsalt = (e*rho/g*hsalt) (ustar^2-ustarT^2)$$

in which e = 1/(3.25u*) is the saltation efficiency expressed as a dimensionless coefficient inversely proportional to the friction velocity and rho is the air density. In the model we have divided qsalt by rho for conversion from kg/m^3 to kg/kg. We have made this clearer in the revised version of the manuscript by specifying the dimensionless (or kg/kg) character of qsalt.

Section 3.3: Can the authors derive any quantification of the sublimation of drifting snow from their simulations? L191: Does the particle absorption of solar radiation increase sublimation? Can such information derived from the model simulations?

You are cordially invited to take a look at the paper submitted to TCD by Le Toumelin et al. (https://tc.copernicus.org/preprints/tc-2020-329/) in which a discussion on sublimation of drifting snow from the simulations presented here is already proposed. To prevent redundancy with that paper and digression from the main objective of the paper that would also not fit with the requirements of model evaluation papers imposed by GMD, we prefer not to focus on that topic here. Moreover, drifting-snow sublimation is the subject of an ongoing paper led by the first author from MAR simulations performed at the scale of the ice sheet, which will be a much better basis to discuss and quantify drifting-snow sublimation than the simulations presented in this paper covering only a small portion of the East Antarctic coast.

Yes, drifting-snow layers in MAR are considered as near-surface clouds and treated accordingly so they indeed contribute to the radiative atmospheric budget. Quantifying the influence of this process on sublimation could be done through sensitivity experiments, for instance by investigating the difference in cloud radiative effect within drifting-snow layers between two model runs in which the drifting-snow scheme is respectively switched on and off. See Le Toumelin et al. (2020) for more details on the radiative effects of blowing snow derived from MAR simulations.

L206-210, and L249-250: What is the rationale for restricting erosion to the surface layer only? The original approach in MAR intuitively makes more sense, where the firn can erode until the mass flux is satisfied, or the snow is too dense/bonded to be erosion.

The original approach in MAR used to work actually differently from what is suggested here by the reviewer. Instead of removing mass layers after layers until the snow mass flux is satisfied, the eroded mass (estimated from the properties of the surface layer only) was then distributed downward among the surface layer as well as all the internal snowpack layers determined as mobile from their current properties (that would have individually led to a different mass to erode if they have been considered as the surface layer) with a decreasing proportion with depth, and removed simultaneously from all these layers, though not in contact with the atmosphere (See Gallée et al., 2001 - Sect. 2.2, P5-6). We have disabled this parameterisation under the consideration that only the surface snowpack layer can exchange momentum and mass with the atmosphere (which is now specified as is in the text), and we have restricted erosion to the surface layer mainly for reasons of numerical stability and computational efficiency (see our more detailed response to comment #6 by reviewer RC1). After obtaining a good agreement between modelled and observed drifting snow mass flux with this new criterion, we have considered it as acceptable.

L254: At item 3: maybe explicitly discuss here the scenario that ER is positive (erosion) *and* the scenario that ER is negative (i.e., deposition).
ER is an erosion rate and is thus always >= 0. This is now specified at item 3.
Deposition of snow (from snowfall and/or deposition of relocated snow) is computed at step 5. To improve clarity, and also following a recommendation made by reviewer RC1, we have reformulated the description of item 5 as: "*The drift fraction is obtained from Eq. (8). Snow is deposited at the surface and surface density is adjusted according to Eq. (7).*".

L285: "further inland" is rather qualitative. Maybe add how many kilometers inland is meant here.
As the exact position of the transition from negative to positive net accumulation along the transect can vary from year to year (see Fig. R2 in our response to comment #10 by reviewer RC1), we have corrected for "*a few kilometers inland*".

L353: It's a little bit strangely formulated, since Fig. 3 only shows D17, not D47. So Fig. 3 is not really showing that the values are closer to observation than for D47.
We have moved the reference to Fig. 3 earlier in the sentence so the new sentence writes: "*The general underestimation in near-surface wind speed at D47 could be caused by the temperature-dependent parameterisation of z0, locally still yielding too high values, while at D17 Fig. 3 illustrates that modelled z0 values are closer to observations.*".

L401-402: I don't comprehend how occurrences are missed at coarser temporal resolution. I assume that the coarser temporal resolution sums the mass fluxes over the coarser time steps, such that no information is lost?
Drifting snow is assumed to occur when the snow mass flux is above 1 g/m/s2. This threshold is given valid for, and used at, a half-hourly resolution. As monthly frequency values are computed from the ratio of half-hourly drifting-snow occurrences in a month over the total number of half-hourly occurrences in that month, similar monthly frequency values could be obtained from different combinations of false negatives compensating false positives within a monthly interval (for instance by overestimating the duration of some events while other are missed).
We have reformulated the sentence which thus becomes: "*MAR shows better results (higher POD and RI) at D17 than at D47, but also simulates more unobserved occurrences (higher FAR) that compensate for missed occurrences in the calculation of monthly frequency values.*".

L380-382: When the duration of events is underestimated, one would also expect an underestimation of total mass flux in events. It seems a bit in contradiction with what is argued later (L416/417) that the main events are correctly simulated and that the underestimation stems from particularly the low wind speed events. I actually think that there is also quite some uncertainty from the simulated firn properties, as mentioned in L428/429.
We have slightly modified that part of the paper to put more emphasis on the possible influence of the misrepresentation of surface snow properties and their temporal evolution, using the comment already

made on possibly exaggerated surface compaction rate as a concrete example. Starting from L414 in the revised version, the paragraph now writes: "*Nearly consistent underestimation of drifting-snow frequency at D47 could also be caused by a misrepresentation of surface snow properties and their temporal evolution. For instance, surface compaction could be locally too strong in the model [...]*".

Fig. 9: It could be a nice addition to show the elevation or terrain slope angle along the transect here as well. It looks like that the terrain gets steeper near the coast and may also exhibit more variability. That variability probably drives SMB variability (as for example shown in Dattler et al. (2019)).
Thank you for this relevant addition. The terrain elevation along the transect is now part of Fig. 9 and the linkages between variability in SMB, erosion and slope are commented in the text: "*MAR represents the general variability in SMB with a strong increase over the first tens of kilometers from the coast and less variability further inland. The variability in i_E/D is more pronounced where the terrain is steeper near the coast and exhibits more variability in topographic surface slope, suggesting that SMB variability is driven by drifting snow. The mean SMB bias is negative [...]*".

L588: "Both parameterizations are given for rho_s = 450 kg/m^3" I don't comprehend this sentence, since the functions B1-B3 are all using a variable rho_s?
This wording was indeed confusing. We changed it for: "*Both parameterisations are given as valid for rho_s values up to 450 kg/m^3.*".

Technical corrections:
- Two comments to make the abstract better comprehensible:

L6: I suggest "drifting-snow compaction of the uppermost firn layer."
Done.

L7/8: I suggest "and a rewrite of the parameterization for the threshold friction velocity, above which snow erosion initiates".
Done.

- L15/16: I suggest "wind-driven ablation or accumulation", since that's in better line with the discussion in the first paragraph.
Done.

- Fig. 1: the red labels on purple background are very difficult to see, and definitely not easy for people with eye-sight problems / color blindness. Maybe put a white box behind the label, or improve the figure otherwise.
This is very true, thanks. We have put a white box behind the station labels to improve readability.

- L134: word missing after "lowest model".
Corrected.

- L155: even though pretty obvious, I recommend to add the value taken for gravitational acceleration.
Done.

- L239: I suggest explicitly referring to Eq. 1.
Done.

- L397: "an estimation"
Corrected.

- L475: "Improvements ... are illustrated"
Corrected.

- Fig. 7: is the horizontal axis the observed or simulated wind speed?
Good catch, thank you. It is the observed wind speed. This is now indicated in the text and in the figure caption.

- I suggest to incorporate Appendix A in the main text.
Done.

References

Amory, C., Trouvilliez, A., Gallée, H., Favier, V., Naaim-Bouvet, F., Genthon, C., Agosta, C., Piard, L., and Bellot, H.: Compari-son between observed and simulated aeolian snow mass fluxes in Adélie Land, East Antarctica, The Cryosphere, 37, 1,373–1,383, https://doi.org/10.5194/tc-9-1373-2015, 2015.

Dattler, M. E., Lenaerts, J. T. M., and Medley, B.: Significant Spatial Variability in Radar-Derived West Antarctic Accumulation Linked to Surface Winds and Topography, Geophysical Research Letters, 46, 13,126–13,134, https://doi.org/10.1029/2019GL085363, 2019.

Le Toumelin, L., Amory, C., Favier, V., Kittel, C., Hofer, S., Fettweis, X., Gallée, H., and Kayetha, V.: Sensitivity of the surface energy budget to drifting snow as simulated by MAR in coastal Adelie Land, Antarctica, The Cryosphere Discuss. [preprint], https://doi.org/10.5194/tc-2020-329, in review, 2020.

Liston, G. E., Haehnel, R. B., Sturm, M., Hiemstra, C. A., Berezovskaya, S., and Tabler, R. D.: Simulating complex snow distributions inwindy environments using SnowTran-3D, Journal of Glaciology, 53, 241–256, https://doi.org/10.3189/172756507782202865, 2007.

Lenaerts, J. T. M., van den Broeke, M. R., Déry, S. J., van Meijgaard, E., van de Berg, W. J., Palm, S. P., and Sanz Rodrigo, J.: Modeling drifting snow in Antarctica with a regional climate model: 1. Methods and model evaluation, J. Geophys. Res.: Atmospheres, 117, https://doi.org/10.1029/2011JD016145, 2012.

Lenaerts, J. T. M., Smeets, C. J. P. P., Nishimura, K., Eijkelboom, M., Boot, W., van den Broeke, M. R., and van de Berg, W. J.:Drifting snow measurements on the Greenland Ice Sheet and their application for model evaluation, The Cryosphere, 8, 801–814, https://doi.org/10.5194/tc-8-801-2014, 2014.

Gallée, H., Guyomarc'h, G., and Brun, E.: Impact Of Snow Drift On The Antarctic Ice Sheet Surface Mass Balance: Possible Sensitivity ToSnow-Surface Properties, Boundary-Layer Meteorology, 99, 1–19, https://doi.org/10.1023/A:1018776422809, 2001.

Gallée, H., Trouvilliez, A., Agosta, C., Genthon, C., Favier, V., and Naaim-Bouvet, F.: Transport of Snow by the Wind: A ComparisonBetween Observations in Adélie Land, Antarctica, and Simulations Made with the Regional Climate Model MAR, Boundary-LayerMeteorology, 146, 133–147, https://doi.org/10.1007/s10546-012-9764-z, 2013.

Genthon, C., Berne, A., Grazioli, J., Durán Alarcón, C., Praz, C., and Boudevillain, B.: Precipitation at Dumont d'Urville, Adélie Land, East Antarctica: the APRES3 field campaigns dataset, Earth Syst. Sci. Data, 10, 1605–1612, https://doi.org/10.5194/essd-10-1605-2018, 2018.

Picard, G., Arnaud, L., Caneill, R., Lefebvre, E., and Lamare, M.: Observation of the process of snow accumulation on the Antarctic Plateau by time lapse laser scanning, The Cryosphere, 13, 1983–1999, https://doi.org/10.5194/tc-13-1983-2019, 2019.

Pomeroy, J. W.: A process-based model of snow drifting, Annals of Glaciology, 13, 237–240, https://doi.org/10.3189/S0260305500007965,1989.

Souverijns, N., Gossart, A., Lhermitte, S., Gorodetskaya, I. V., Grazioli, J., Berne, A., Duran-Alarcon, C., Boudevillain, B., Genthon, C., Scarchilli, C., and van Lipzig, N. P. M.: Evaluation of the CloudSat surface snowfall product over Antarctica using ground-based precipitation radars, The Cryosphere, 12, 3775–3789, https://doi.org/10.5194/tc-12-3775-2018, 2018.

Vionnet, V., Brun, E., Morin, S., Boone, A., Faroux, S., Moigne, P. L., Martin, E., and Willemet, J. M.: The detailed snowpack scheme Crocusand its implementation in SURFEX v7.2, Geosci. Model Dev., 5, 773–791, https://doi.org/10.5194/gmd-5-773-2012, 2012.

van Wessem, J. M., van de Berg, W. J., Noël, B. P. Y., van Meijgaard, E., Amory, C., Birnbaum, G., Jakobs, C. L., Krüger, K., Lenaerts, J.T. M., Lhermitte, S., Ligtenberg, S. R. M., Medley, B., Reijmer, C. H., van Tricht, K., Trusel, L. D., van Ulft, L. H., Wouters, B., Wuite, J., and van den Broeke, M. R.: Modelling the climate and surface mass balance of polar ice sheets using RACMO2 – Part 2: Antarctica(1979–2016), The Cryosphere, 12, 1,479–1,498, https://doi.org/10.5194/tc-12-1479-2018, 2018.